# ConceptFactory: Facilitate 3D Object Knowledge Annotation with Object Conceptualization

**Jianhua Sun**[†], **Yuxuan Li**[†], **Longfei Xu**[‡], **Nange Wang**[‡], **Jiude Wei**[‡], **Yining Zhang, Cewu Lu**[§]
Shanghai Jiao Tong University

## Abstract

We present **ConceptFactory**, a novel scope to facilitate more efficient annotation of 3D object knowledge by recognizing 3D objects through generalized concepts (*i.e.* object conceptualization), aiming at promoting machine intelligence to learn comprehensive object knowledge from both vision and robotics aspects. This idea originates from the findings in human cognition research that the perceptual recognition of objects can be explained as a process of arranging generalized geometric components (*e.g.* cuboids and cylinders). ConceptFactory consists of two critical parts: i) **ConceptFactory Suite**, a unified toolbox that adopts Standard Concept Template Library (STL-C) to drive a web-based platform for object conceptualization, and ii) **ConceptFactory Asset**, a large collection of conceptualized objects acquired using ConceptFactory suite. Our approach enables researchers to effortlessly acquire or customize extensive varieties of object knowledge to comprehensively study different object understanding tasks. We validate our idea on a wide range of benchmark tasks from both vision and robotics aspects with state-of-the-art algorithms, demonstrating the high quality and versatility of annotations provided by our approach. Our website is available at https://apeirony.github.io/ConceptFactory.

## 1 Introduction

In the current data-driven era, the availability of a large amount of training data with dense annotations has become an indispensable factor for the successful implementation of deep neural networks in a wide range of 3D object understanding tasks. Particularly, for tasks like segmentation, pose estimation and more sophisticated robot manipulation, current approaches [1, 2, 3, 4, 5, 6, 7, 8] require a substantial volume of annotations of semantic, pose and affordance knowledge to fully demonstrate their power.

However, there are two primary issues demanding attention in 3D object knowledge annotation. On one hand, some types of knowledge such as affordance for manipulation are highly complicated to manually annotate [7], resulting in few existing datasets being available for such labels. On the other hand, common practices of acquiring these knowledge annotations [6, 9, 10] follow the conventional paradigm that only a single type of knowledge is labeled on one object at a time, for which researchers develop different annotation platforms to adapt to various knowledge types and let annotators engage in multiple rounds of annotations, taking significant time and human effort.

In this paper, We present **ConceptFactory** as a novel annotation paradigm that addresses these existing issues and facilitates more efficient annotation of 3D object knowledge. The idea behind ConceptFactory originates from the well-known 'Recognition-by-Components' theory [11] in human cognition research, which finds that the perceptual recognition of objects can be explained as a process of arranging generalized geometric components. Inspired by this theory, we devise an efficient knowledge annotation paradigm performing in two steps. i) Describe the shape of an object

---

[†]and [‡] denote equal contribution, [§] denotes corresponding author

38th Conference on Neural Information Processing Systems (NeurIPS 2024) Track on Datasets and Benchmarks.

with generalized geometric concepts, or in other words, object conceptualization. ii) Procedurally define (different types of) knowledge on these generalized concepts. In this manner, all types of knowledge defined on the concepts can be automatically propagated to the object as various types of annotations, taking advantage of correspondence between the concepts and the object shape.

ConceptFactory provides a favorable solution to both aforementioned issues. First, manual knowledge annotation on 3D objects, which can be very complicated in some cases, is no longer required. Instead, researchers only need to procedurally define a type of knowledge with mathematical rules on certain concepts, and these knowledge will be automatically propagated to all target objects consisting of such concepts. Second, intensive human effort is required only once during object conceptualization, compared to the conventional annotation paradigm where significant labor and time resources are repeatedly expended for annotating each type of knowledge.

ConceptFactory comes with two critical components. The first one is **ConceptFactory Suite**, a unified toolbox that adopts Standard Concept Template Library (STL-C) to drive a web-based platform for object conceptualization. The STL-C consists of 263 concept templates that comprehensively covers the essential structure of daily objects, and the conceptualization platform guides users to select and parameterize concept templates in STL-C to describe a given object and thereby obtains the conceptualization result. Then, a wide range of knowledge, which is procedurally defined on the templates, can be automatically propagated to the object as annotations. The other component is **ConceptFactory Asset**, a large collection of conceptualized objects acquired using ConceptFactory suite, containing 4380 objects from 39 categories involving 39k template instances and 295k parameters. We present such asset considering that the object conceptualization process still requires certain human effort, thereby offering already conceptualized objects to the community would make it convenient for researchers to use and study on, *e.g.* customizing their own knowledge and conduct experiments with them.

The knowledge annotations offered by our approach are mathematically grounded and functionally aligned, serving as a catalyst for machine intelligence to recognize and interact with objects. We demonstrate the effectiveness of our idea from both vision and robotic aspects on a wide range of benchmark tasks including segmentation, pose estimation and robot manipulation through state-of-the-art algorithms, figuring out that our approach can easily gather various types of annotations, with quality comparable or even better than those acquired through conventional annotation paradigms.

## 2  Related Works

### 2.1  Object Recognition in Human Cognition

Over the last few decades, numerous studies on cognitive science [11, 12, 13, 14, 15] have placed their focus on the inner mechanisms within human perception of objects, and consider conceptual knowledge having major influences during such process [11, 14, 16, 17]. Biederman [11] found that the perceptual recognition of objects is conceptualized to be a process in which an object is segmented into an arrangement of simple geometric components, such as blocks, cylinders, wedges, and cones. Meanwhile, other studies [14, 18, 19] also indicates a strong connection between human perception and conceptual knowledge. Such connection is even stronger for infants [20, 21], since they are way less susceptible to empirical influences. These findings reveal a plausible path for human understanding of objects, and also inspire us with a novel methodology to label abundant human knowledge on objects, thereby helping intelligent agents to better understand the physical world.

### 2.2  3D Object Understanding Tasks

As the basic elements that constitute our daily life, 3D objects usually carry abundant information within their physical shapes assigned by humans. Given such crucial status of 3D objects, it is of great importance to teach machine intelligence to understand them and thereby enable it to perceive and interact with the objects. This involves both vision and robotics aspects. For vision tasks, one of the frequently studied task is part segmentation [1, 2, 6, 10, 22, 23, 24], which aims at assigning various types of pre-defined labels to points on the object. Additionally, some recent studies also focus on part pose estimation [6, 24, 25], which queries the 6-dimensional transformation of detected parts on the object, inferring their scales, rotations and positions. For robotics tasks, many studies

focus on interacting with articulated objects [6, 7, 8, 24, 26, 27], and refer to interaction success rates as a measurement of performance.

## 2.3 3D Object Datasets

Throughout the years, datasets paved the way for machine learning across various modalities [28, 29, 30, 31, 32], empowering neural networks to carry out numerous sophisticated tasks [1, 33, 34, 35, 36, 37]. However, regarding the perception and interaction with 3D objects which play an important role in daily life, most of the related large-scale object datasets throughout the years [32, 38, 39, 40, 41] only provide class labels for each model, making them suitable for a very limited variety of tasks like classification. In order to mitigate this issue, many researchers have placed their efforts on annotating more detailed knowledge onto 3D models in these object datasets. For example, several existing popular datasets are derived from ShapeNet [32]. ShapeNetPart [42] offers part-level semantic segmentations of the models across 16 categories, and PartNet [9] goes one step further and provides fine-grained segmentations across 24 categories. More recently, PartNet-Mobility [10] was proposed with URDF styled annotations, which adds joint information for articulated objects. And GAPartNet [6] offers annotations on generalizable and actionable parts (GAParts) which share similar functionalities across different object categories. These valuable contributions significantly facilitate a wide range of both vision and robotics tasks.

## 2.4 Knowledge Acquisition on 3D Objects

Apart from gathering 3D object assets, it is also crucial to align various types of knowledge onto these objects to enable training for modern-day networks. However, such knowledge are typically acquired under type-specific paradigms. For example, knowledge like segmentations are acquired by either 'painting' 2D projections [42] or splitting/merging meshes [6, 9], whereas pose-related knowledge are generally acquired via oriented bounding boxes [6]. Affordance knowledge, being much less definitive and task-specific, is very difficult to collect human annotations [7]. Instead, one line of work [7, 8, 26] use repetitive random agent trials in simulations to acquire actionability scores over pixels on object surfaces by observing the state changes of the object after each trial. As the diverse array of knowledge types requires different annotation manners, it can be labor-intensive and time-consuming to acquire a full set of knowledge for an object. In this paper, we resolve this issue by proposing an efficient universal knowledge acquisition paradigm based on conceptualization.

# 3 ConceptFactory Suite

We develop the ConceptFactory suite as a general toolbox for object conceptualization. It consists of two major components, namely the Standard Concept Template Library (STL-C) and the corresponding conceptualization platform. We will introduce the construction of STL-C in Sec. 3.2. Then in Sec. 3.3 we delve into the conceptualization platform and demonstrate how to use STL-C to describe an object. Finally, we show how these conceptual descriptions facilitate efficient object knowledge annotation in Sec. 3.4. But first, we discuss the motivations of developing ConceptFactory in Sec. 3.1.

## 3.1 Why Concepts?

The entire ConceptFactory suite takes inspiration from the advancements of researches on human cognition and brain science [11, 12, 13, 14, 15, 16], where it is discovered that we humans learn about the physical world by perceiving geometry patterns from objects and inducing them along with related knowledge as commonsense for future reference. Based on such findings, we establish a novel knowledge annotation paradigm for object understanding tasks by explicitly modelling such abstract commonsense information as concepts for regular geometry patterns and reversing the induction process. Specifically, by generalizing the concepts towards certain objects, various knowledge associated with the concepts can be automatically propagated to all these objects. An illustration of such process is shown in Fig. 1-Left. Compared with conventional annotation process where only one object is labeled with a single type of knowledge at a time, such evolution in annotation paradigm will greatly speed up the knowledge annotation process as well as diversify the types of knowledge that can be annotated to the objects, empowering more sophisticated tasks in the data-driven era.

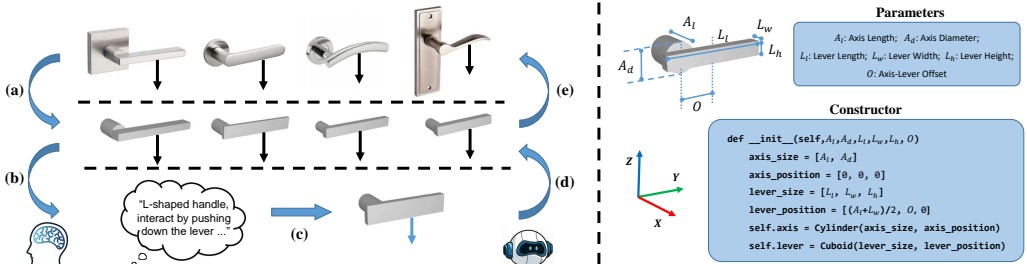

Figure 1: **[Left]** Illustration of the relationship between human cognition (a-b) and our approach (c-e), exemplified by handle as object and affordable interaction as knowledge. (a) Human recognizes objects as an arrangement of geometric components. (b) Abstract commonsense information are induced from the geometries in human mind. (c) Explicitly model the abstract information as a regular geometry concept with specific knowledge. (d) Generalize the concept towards different objects. (e) Propagate the knowledge from the concept to objects as annotations. **[Right]** Example of parameters and the constructor of a concept template. Please refer to the codes in our website for concept template implementations.

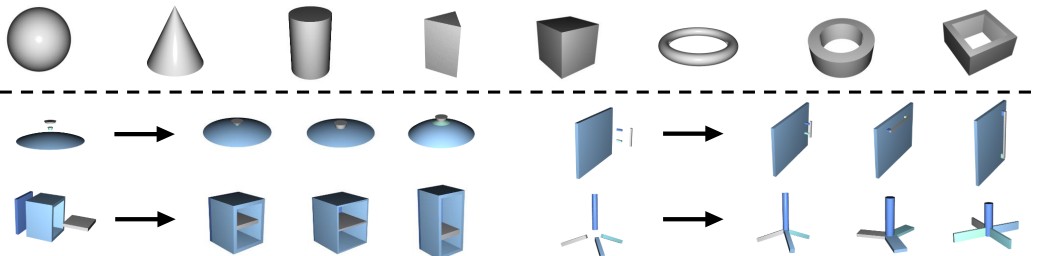

Figure 2: Shape instances of geometry (Top) and concept (Bottom) templates with specific parameters. **[Bottom]** The figures on the left side of the arrows display each geometry component of a concept template individually, whereas those on the right side are example instances of concept templates with various parameters. The instance at bottom-right is the result of modifying discrete parameters.

## 3.2   Standard Concept Template Library

Named after STL in C++ which provides commonly used program templates, our Standard Concept Template Library, *a.k.a.* STL-C, consists of templates for numerous generalized geometry patterns that are frequently notable in daily life. Each of the concept templates is implemented as a Python class template, and can be instantiated as a concept instance that describes a 3D shape when given the template parameters. A brief illustration of the template architecture is in Fig. 1-Right.

**Geometry Templates.**   To avoid the tedious effort of repeatedly defining frequently used geometries from scratch when developing concept templates, a good solution is to first build templates of these geometries aiming at facilitating the construction of concept templates through inheritance. To achieve this goal, we parameterize the geometries by introducing geometry templates, which can be instantiated into various geometry instances given different parameters. In practice, the construction of STL-C has involved ten geometry templates, with some frequently used ones shown in Fig. 2-Top.

**Concept Templates.**   Based on geometry templates, we can easily construct concept templates as a descriptor of geometry patterns. Specifically, each of the concept templates explicitly depicts a geometry pattern by assembling various templates* under specific constraints embedded in the pattern. Such constraints will manifest during parameterization of the concept template. That is, the parameters for the concept template will be processed under constraint-defined rules to generate parameters for member geometry templates, which then instantiate accordingly as parts of the concept instance. Particularly, when describing periodic patterns, we introduce additional discrete parameters

---

*Both geometry templates and concept templates are applicable.

to concept templates to specify the number of repetitions in geometry instances. Fig. 2-Bottom shows examples of some concept templates.

**Discussion.** Currently, we have included a total of 263 distinct concept templates into STL-C and found their capability to comprehensively cover a total of 4380 objects across 39 categories. In addition to the concept templates available in STL-C, it is also worth noting that thanks to the inheritable nature of a template representation, users can easily customize new templates using existing ones to cover novel shapes in specific applications. Such property of STL-C will significantly improve its representational power.

## 3.3 Conceptualization Platform

With STL-C properly constructed, we continue to discuss the detailed steps for utilizing the library to conceptualize 3D objects, *i.e.* describe the objects with generalized concepts. Specifically, we first discuss the principles for describing objects using STL-C in Sec. 3.3.1. Then we develop a web-based conceptualization platform as a tool for acquiring conceptualization results in Sec. 3.3.2.

### 3.3.1 Principles for Object Conceptualization with STL-C

Compared with the complexity of an object as a whole, each of the object's parts typically enjoys a much simpler structure as well as less variations, making them more suitable as units for conceptualization. Therefore, We divide objects of each category into a group of parts according to their structural hierarchy for better guidance to the conceptualization process. In practice, for each of the object's parts, we first choose a template from STL-C whose embedded concept best matches the part's geometric structure, and then we carefully parameterize the template so that the resulting concept instance serves as an effective approximation of the part. These parameterized template instances are then spatially arranged through parameterized spatial transformations so that they are aligned with the object's respective parts, forming an conceptualization of the object.

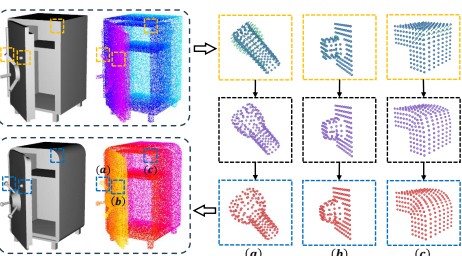

Figure 3: **[Left]** Minor gaps in geometric details between the original object (bottom) and its conceptualization (top). **[Right]** Restoring the geometric details via deformation based on point-wise correspondences.

As the conceptualization is capable of effectively representing the object's structure in general, there remains a gap between the object's actual shape and the structure in terms of geometric details, see Fig. 3-Left. Such gaps are typically the consequences of objects having local shape irregularities. We draw inspiration from BPS [43] and address this issue by establishing a point-wise correspondence between the concepts and the actual shape of the object. Specifically, for each point $\mathbf{x}$ on the object surface, we find a corresponding point $\mathbf{y}$ on the concept instance that minimizes $L_2(\mathbf{x}, \mathbf{y})$. By establishing such correspondence, we can restore the geometric details for the conceptual description of an object by applying deformation $(\mathbf{x} - \mathbf{y})$ to $\mathbf{y}$, as is illustrated in Fig. 3-Right.

### 3.3.2 Conceptualization Platform

**Web-based Interface.** To efficiently perform the conceptualization process, we devise a web-based interface which divides the whole process into specific user tasks. Through such system, users can easily choose concept templates for the parts of a given object, and adjust their parameters for optimal approximation. Both the target object and the parameterized concept instances are rendered in real-time as reference. Fig. 4 gives an overview to our conceptualization interface and workflow.

**Concept Parameter Optimizer.** To further speed up the conceptualization process for each of the object's parts, we introduce a template parameter optimizer to the platform that is capable of automatically adjusting the concept parameters with a single click. The optimizer is made possible thanks to the compatibility of concept templates to differentiable rendering. Specifically, a concept instance can be rendered through 1) differentiable calculations for parameters of member geometry templates, and 2) differentiable deformations on the geometry templates' respective default template

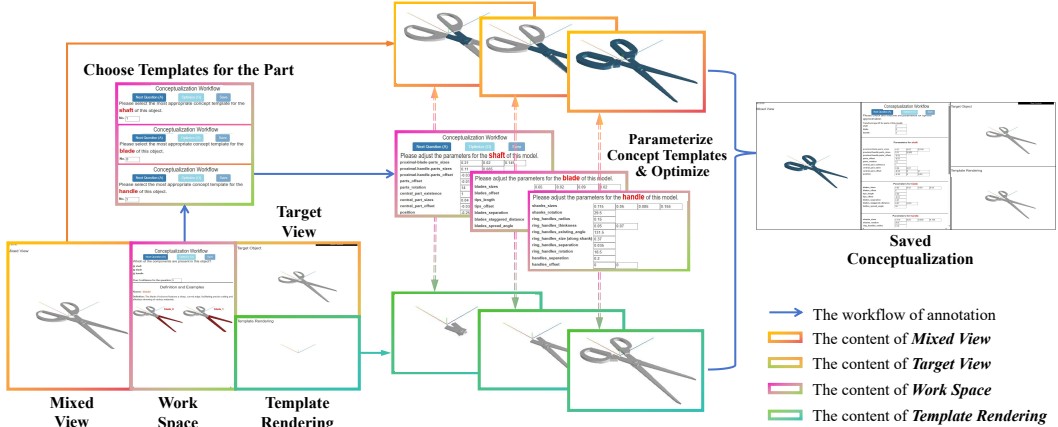

Figure 4: An overview to our conceptualization interface and the workflow (blue arrow). The interface is divided into four components: work space, target view, template rendering, and mixed view. In work space, users first select best-match templates for each part of the target object, then parameterize each concept template with the help of the optimizer, and finally save the conceptualization result. Target view illustrates the shape of the target object, template rendering displays instances of concept templates with current parameters, while mixed view visualizes the integration between target view (gray) and template rendering (blue), helping users perform the conceptualization efficiently. Zoom in for a clear view.

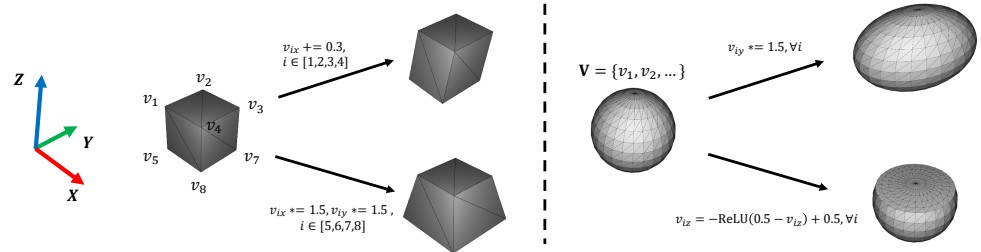

Figure 5: Examples of differentiable deformations on template instances with default parameters. **[Left]** A default quadrangular prism instance in mesh with all edge lengths set to 1, bearing the same shape as a cuboid. Its eight vertices are labelled from $v_1$ to $v_8$. The upper case show a translation is applied to four top vertices for a 3d parallelogram, and the lower case show a scaling is applied to four bottom vertices for a frustum of a pyramid. **[Right]** A sphere mesh with radius one and set of vertices $\mathbf{V}$. The upper case show a scaling is applied to $y$-axis of all vertices for an ellipsoid, and the lower case show we use ReLU operation to truncate a sphere. Through differentiable transformations (addition, multiplication, *etc.*) on their respective vertices, the shapes can be deformed in a differentiable manner.

instances *(a template instance with default parameters, e.g. a sphere with radius 1)* parameterized by the instances' parameters. Fig. 5 provides more details. By calculating and minimizing the gap with loss functions between the template instance and the corresponding object part, the gradients can be directly back-propagated to the template's parameters for updates. We adopt Point2Mesh loss [44] in our implementation. Benefiting from the optimizer, a large number of tedious parameter adjustments can be automatically achieved, and users just need to refine parameters that are not properly optimized. The incorporation of parameter optimizer greatly reduces the workload during conceptualization from about 10 min to 7 min per object on average.

### 3.4 Procedural Knowledge Annotation

After obtaining the conceptual description of an object, we explain how different types of knowledge are annotated on the object. A significant benefit of our concept templates is its compatibility to procedural definitions of knowledge, facilitating a fully automatic knowledge annotation scheme.

Particularly, by implementing mathematically defined knowledge as attributes of STL-C templates, the knowledge on template instances can be propagated to the object, according to correspondence between the concepts and the object shape built during conceptualization. Considering the common knowledge types on objects can be broadly classified into two distinct categories, namely region-based knowledge and pose-based knowledge, we introduce their annotation mechanisms respectively as follows. A brief illustration is in Fig. 6.

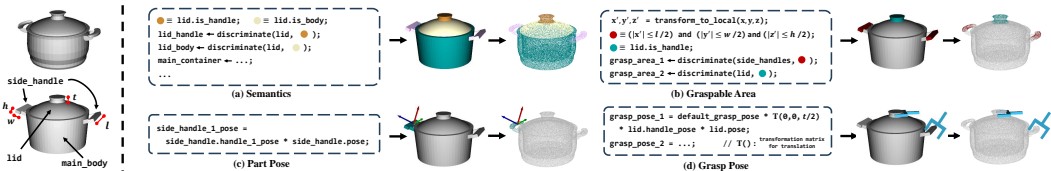

Figure 6: **[Left]** Conceptualization results of a KitchenPot, geometric details and certain parameters are omitted for simplicity. **[Right]** Procedural annotation for different types of knowledge. (a-b) Region-based knowledge like semantics and affordable area is implemented through region discrimination function. (c-d) Pose-based knowledge like part pose and grasp pose is implemented with transformations from local to world coordinates. Please refer to the codes in our website for detailed implementations.

**Region-Based Knowledge Annotation.** We consider region-based knowledge as collections of labeled regions on an object, with one most common example being semantic segmentation. To enable region-based knowledge annotation, users can implement region descriptions as a region discrimination function which decides whether a given point is located within the designated region of the concept instance. Then the set of points within this region can be considered as conforming to such knowledge. In this manner, numerous region-based knowledge can be easily defined onto the templates. And further through the point-wise correspondence (Sec. 3.3.1), the annotation of each point on the object can be obtained at a fine-grained level, by propagating the knowledge from its corresponding point on the concept.

**Pose-Based Knowledge Annotation.** For pose-based knowledge such as part pose, grasp pose, *etc.*, they can be initially defined in a concept's local coordinates, and then gradually transformed to world coordinates as the concept finds its place in the overall object description.

**Visualization of Knowledge Annotation.** We present visualizations of various knowledge types (affordance, semantic segmentation, and part pose) across object categories in Fig 7, showing comparisons between original annotations (Left) and those from ConceptFactory Suite (Right). Object **affordances** are highlighted in red regions. Compared to the conventional annotation method [7], our approach enjoys two important benefits. First, our approach provides more accurate and consistent labels with less noise, which greatly facilitates manipulation frameworks to better learn object affordance knowledge and thereby enhances its power. Second, the affordance labels are assigned by human experts instead of being acquired by trials in simulation environments [7]. This ensures controllable manipulation as robots just learn those kind of affordance that provided by human preferences. **Semantic segmentation** annotations are denoted by distinct colors, and our approach produces annotations nearly identical to the original ones, or even better in some cases. *e.g.* for *chair* in Row 1, Col 3-4, the original annotations confuse with the definition of crossbars between legs while our approach reasonably and consistently labels these regions to part of legs. **Part poses** are denoted by oriented bounding boxes, and our method accurately matches the precision of the original annotations.

**Discussion.** Compared with the conventional knowledge annotation paradigm where the annotation process is performed on one object at a time for every type of knowledge (*e.g.* about 8 min/obj for part semantics [9] and 10 min/obj for part pose [6]), the concept based object description offers a once-and-for-all alternative at a much lower cost. Specifically, by defining knowledge on relevant concept templates, the corresponding knowledge annotation is automatically completed once conceptualization is performed on the object. The human time cost for our approach only involves object conceptualization, which is about 7 min/obj. The comparison demonstrates that, as the number

of knowledge types requiring annotation increases, the superior efficiency of our approach gradually becomes more evident.

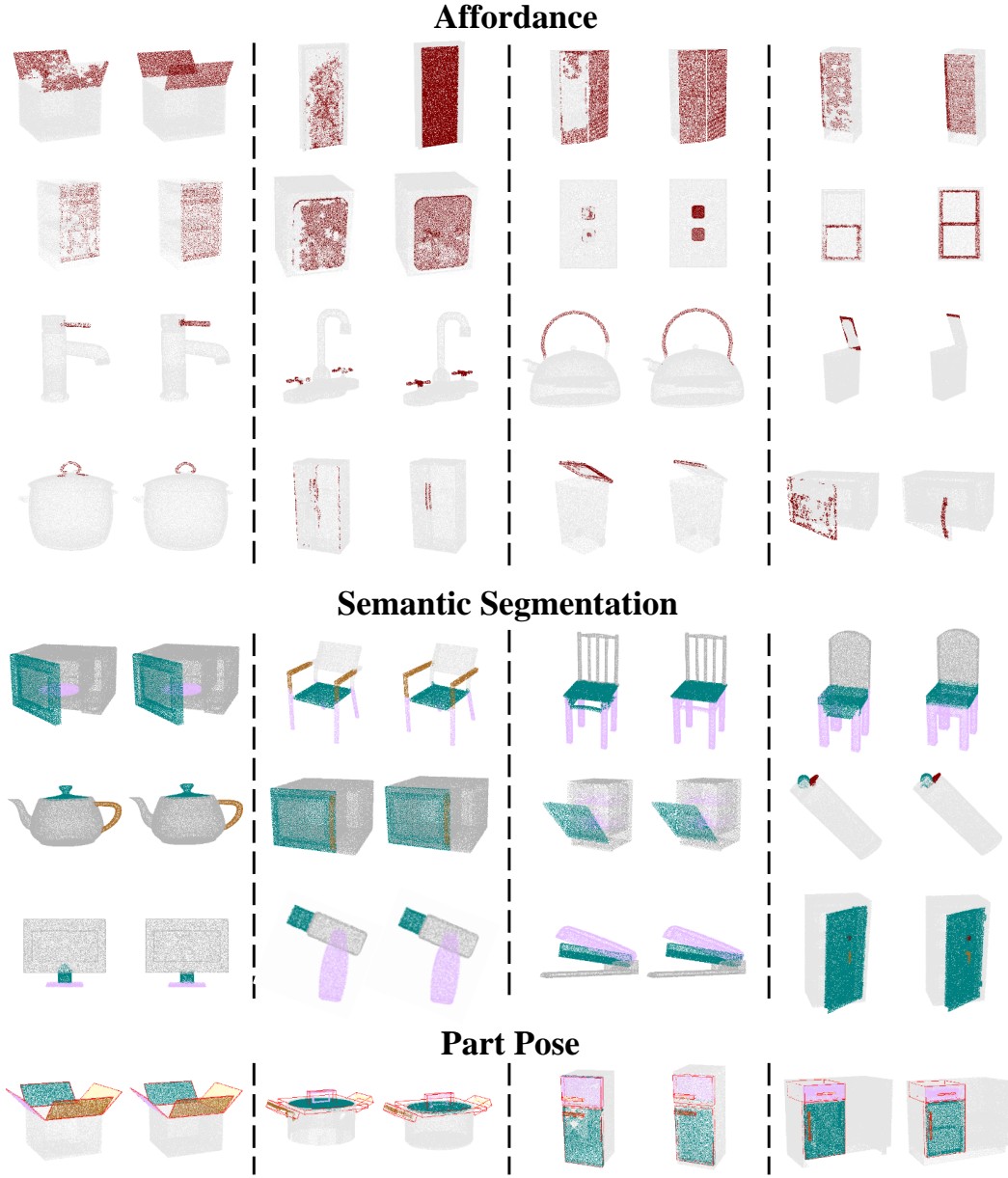

Figure 7: Visualization of different types of knowledge annotations including affordance (Push - Row. 1-3, Pull - Row. 4-5), semantic segmentation and part pose. **[Left]** Annotations acquired by conventional approaches [6, 7, 45]. **[Right]** Annotations acquired by our approach.

## 4    ConceptFactory Asset

With the help of ConceptFactory suite, we further present a large-scale ConceptFactory asset, providing fine-grained conceptualization results for 4380 objects from 39 categories with 39k geometry instances and 295k parameters, including an average of 8.8 geometry instances and 67.4 parameters in each conceptualization result. We select objects from popular sources [9, 10, 32, 42, 46] that i)

are widely used in vision and manipulation tasks, ii) include both CAD and scanned models, and iii) yield rich and diverse conceptualization results. Tab. 1 shows the statistics. These assets enable the community to avoid certain human efforts involved in object conceptualization. Our purpose of releasing these assets is to help researchers easily acquire a large amount of well-annotated data to meet their research needs in specific applications, by customizing knowledge according to Sec. 3.4.

|           | Bot  | Box  | Bkt  | Chr  | Clp  | Dsw  | Dsp  | Dpl  | Dor  | Drh | Egl  | Fct  | Fdr | Glb  | Gsk | Ket  | Ktp  | Knf  | Ltp  | Lgt  |
|-----------|------|------|------|------|------|------|------|------|------|-----|------|------|-----|------|-----|------|------|------|------|------|
| $N$       | 446  | 98   | 54   | 746  | 49   | 185  | 86   | 262  | 28   | 3   | 119  | 250  | 8   | 38   | 44  | 24   | 24   | 76   | 48   | 60   |
| $I_{ttl}$ | 2.2k | 374  | 235  | 7.2k | 236  | 1.2k | 602  | 971  | 269  | 9   | 1.2k | 1.8k | 46  | 245  | 132 | 261  | 227  | 365  | 146  | 381  |
| $I_{med}$ | 5    | 4    | 5    | 9    | 4    | 5    | 7    | 4    | 10   | 3   | 9    | 7    | 5   | 6    | 3   | 11   | 10   | 5    | 3    | 4    |
| $I_{max}$ | 6    | 7    | 6    | 36   | 6    | 19   | 10   | 7    | 19   | 3   | 17   | 20   | 7   | 15   | 3   | 14   | 13   | 9    | 5    | 15   |
| $P_{ttl}$ | 11k  | 2.8k | 1.1k | 39k  | 974  | 7.7k | 3.3k | 9.2k | 1.9k | 50  | 5.9k | 12k  | 214 | 1.4k | 965 | 1.9k | 1.2k | 2.2k | 1.7k | 1.7k |
| $P_{med}$ | 28   | 28   | 20   | 51   | 15   | 37   | 41   | 37   | 71.5 | 19  | 54   | 48   | 26  | 38   | 22  | 73   | 50   | 28   | 37   | 26   |
| $P_{max}$ | 28   | 54   | 26   | 80   | 27   | 104  | 41   | 51   | 97   | 19  | 66   | 61   | 28  | 44   | 22  | 105  | 56   | 36   | 41   | 55   |
|           | Mcw  | Mug  | Ovn  | Pen  | Plr  | Rfg  | Rlr  | Saf  | Scs  | Smp | Shv  | Stp  | Stf | Swt  | Tab  | Tcn | USB  | Wsm  | Win  | TTL  |
| $N$       | 76   | 113  | 18   | 112  | 19   | 65   | 15   | 22   | 100  | 46  | 9    | 34   | 637 | 84   | 136  | 124 | 54   | 17   | 51   | 4380 |
| $I_{ttl}$ | 476  | 499  | 429  | 747  | 198  | 556  | 45   | 245  | 1.4k | 228 | 27   | 237  | 12k | 327  | 1.9k | 528 | 394  | 140  | 341  | 39k  |
| $I_{med}$ | 6    | 5    | 22   | 6    | 11   | 8    | 3    | 11   | 13   | 5   | 3    | 6.5  | 15  | 3    | 11   | 4   | 6.5  | 7    | 5    | -    |
| $I_{max}$ | 16   | 8    | 36   | 10   | 13   | 24   | 3    | 17   | 19   | 10  | 3    | 13   | 82  | 25   | 45   | 16  | 14   | 14   | 13   | -    |
| $P_{ttl}$ | 4.2k | 3.1k | 3.3k | 4.4k | 1.2k | 3.7k | 318  | 1.6k | 7.6k | 1.4k| 243  | 1.4k | 120k| 2.4k | 24k  | 3.6k| 1.8k | 1.3k | 3.2k | 295k |
| $P_{med}$ | 58   | 28   | 188  | 34   | 60   | 55   | 20   | 68.5 | 76   | 29  | 27   | 42   | 180 | 27   | 127  | 31  | 35   | 77   | 63   | -    |
| $P_{max}$ | 123  | 39   | 259  | 61   | 70   | 142  | 23   | 131  | 83   | 43  | 27   | 46   | 248 | 48   | 521  | 48  | 38   | 77   | 92   | -    |

Table 1: ConceptFactory asset statistics per object category. Each category is denoted by a 3-character code ('TTL' for 'Total') and see the supplementary material for the cross reference table. For each category, we count the number of objects $N$, as well as the number of geometry instances $I$ and parameters $P$ in the conceptualization results. For $I$ and $P$, we report the total ($ttl$), median ($med$) and maximum ($max$) value. More visualizations and discussions are in the supplemental material.

|          | Semantic Segmentation | | | | Cross Category Segmentation | | Part Pose Estimation | | |
|----------|----------|-------|----------|-------|------|------------|-------|-------|----------|
|          | PointTransformer | | PointNet++ | | | | GAPartNet | | |
|          | mAcc | mIoU | mAcc | mIoU | mAP | mAP$_{50}$ | mIoU | $A_5$ | $A_{10}$ |
| Original | 90.0 | 75.4 | 88.8 | 67.2 | 30.0 | 37.7 | 42.6 | 29.1 | 55.7 |
| ConFac   | 89.8 | 75.6 | 88.8 | 67.3 | 30.6 | 38.5 | 42.8 | 30.3 | 56.9 |
| $\Delta$ | -0.2 | 0.2 | 0.0 | 0.1 | 0.6 | 0.8 | 0.2 | 1.2 | 1.2 |

Table 2: Experiment results on vision tasks. Detailed results are in the supplementary material.

## 5 Experiments

We conduct exhaustive experiments on ConceptFactory assets across various vision and manipulation tasks to validate the effectiveness of our knowledge annotation scheme. Specifically, we train baseline networks twice to get two separate models using either the original annotations, which are obtained according to conventional knowledge acquisition paradigms, or annotations provided by our scheme on the same set of objects. We regard the performance gap between the two models as a measurement for our annotation's quality. See supplementary material for detailed experiment settings and results.

### 5.1 Vision Tasks

**Overview.** We conduct experiments on three vision tasks, *i.e.* semantic segmentation, cross category part segmentation and part pose estimation, to demonstrate the versatility of our concept-based annotation scheme. We choose PointTransformer [2] and PointNet++ [22] as baseline networks for semantic segmentation task, and GAPartNet [6] for the other two tasks. The performance is evaluated on test objects with the original annotations as ground truth. The experiments involve 2316 objects in 29 categories from ConceptFactory assets which possess original annotations for these tasks.

**Main Results.** Tab. 2 reports the performance of two separate baseline models using the original and our annotations for training on three different tasks, showing only minor discrepancies between them. This concretely proves that annotations provided by our scheme possess high quality on par with original ones. Specifically, segmentation and pose estimation task respectively demonstrate the

effectiveness of our approach in providing region-based and pose-based knowledge. We attribute a slight performance decline in some cases to the cognitive variance between our annotators for object conceptualization and the annotators involved in obtaining the original annotations, as we use the original annotations as ground truth to evaluate the performance of both models. For certain performance improvements, we consider the fact that our mathematically grounded annotations can offer better consistency and lower noise, which supervise networks for better convergence.

## 5.2 Manipulation Tasks

**Overview.** We proceed to analyze the superiority of our approach in providing annotations for manipulation tasks. We introduce Where2Act [7], Where2Explore [8] and GAPartNet [6] as baseline approaches, where the former two rely on affordance annotations for training and the latter one relies on pose annotations. Unlike vision tasks where annotations for ground truths are still labeled by human based on one's subjective cognition, performance on manipulation tasks can be fairly evaluated via success rate, a fully objective metric. Therefore, manipulation tasks can serve as a concrete benchmark to more accurately validate the quality of our annotations. Particularly, we select 989 objects in 15 categories from ConceptFactory assets, which are suitable for single-gripper-manipulation. These objects effectively cover a wide range of 26 manipulation tasks. The experiments are conducted in SAPIEN simulator [10].

**Main Results.** For Where2Act and Where2Explore, they require affordance annotations for training. However, it is extremely difficult to collect human annotations with previous annotation scheme [7], and instead they acquire the affordance by interacting with objects in simulation. As our approach allows for human assignment of affordance annotations, we compare the performance of baseline methods trained with the original and our affordance annotations in Tab. 3. Considering affordance forms can vary significantly across different manipulation tasks, the remarkable improvements, up to 26.6% on Where2Act-push, suggest the superiority of our affordance annotations in terms of both versatility and quality. Particularly, we attribute the prominent improvements to three benefits of our annotations compared with original ones acquired in simulation: i) better annotations consistency, ii) less noise, and iii) avoiding inaccurate labels caused by imperfections in simulation. We also regard this capability of providing affordance knowledge as the major indication of the advantages of our annotation scheme.

For pose guided approach GAPartNet, the performances of models trained with the original and our annotations are very similar, further proving the effectiveness of our scheme on part pose annotation.

|  | Where2Act | Where2Explore | GAPartNet |
|---|---|---|---|
| Original | 23.7 / 8.4 | 33.0 / 15.6 | 24.8 |
| ConFac | 30.0 / 10.6 | 37.4 / 18.5 | 25.1 |
| Δ | 6.3 / 2.2 | 4.4 / 2.9 | 0.3 |

Table 3: Success rates on average of manipulation tasks. Enteries for Where2Act and Where2Explore is reported for *push / pull* actions separately following their setting.

## 6 Conclusion

In this paper, we introduce ConceptFactory, a novel scope to facilitate more efficient annotation of 3D object knowledge by recognizing 3D objects through generalized concepts, in order to promote machine intelligence to learn comprehensive knowledge for various 3D object understanding tasks of both vision and robotics aspects. ConceptFactory suite is first proposed as a unified toolbox that adopts Standard Concept Template Library (STL-C) to drive a web-based platform for object conceptualization. Taking advantage of the concept parameter optimizer, users can easily conceptualize an object with the platform and then automatically annotate various types of knowledge on the object in a procedural process. We further present ConceptFactory assets as a large collection of conceptualized objects acquired using ConceptFactory suite. With these assets, researchers can avoid certain human efforts for object conceptualization and expeditiously acquire huge rich-annotated data for their studies. We comprehensively evaluate the effectiveness of our annotation paradigm on four representative object understanding tasks from both vision and robotic aspects, and the experiments suggest the high quality and versatility of annotations provided by our approach.

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
