We provide comprehensive supplementary materials for better understanding of our paper and show more evidence to support our idea. The appendices are organized as follows: Sec. A first provides some discussions for certain points. Then we further provide detailed experiment settings, results, analysis and visualizations in Sec. B. Finally, we show details for STL-C and ConceptFactory asset in Sec. C.

## A Discussions

### A.1 Purpose behind ConceptFactory

In this paper, we present the idea of ConceptFactory to facilitate more efficient annotation of 3D object knowledge by recognizing 3D objects through generalized concepts. We would like to emphasize that our purpose mainly focuses on providing an advanced practice in annotation collection. Although we also provide ConceptFactory asset which is a large collection of conceptualized data, the asset is not designed as a dataset for a specific task. The main purpose of ConceptFactory asset is sharing the fine-grained conceptualization results for a large scale of objects across diverse categories with the community, which can enable the community to avoid certain human efforts involved in object conceptualization and then help researchers easily acquire a large amount of well-annotated data to meet their research needs in specific applications by customizing their own knowledge according to the procedural knowledge annotation scheme.

### A.2 Human-made *v.s.* natural objects

In this paper, we develop ConceptFactory and validate its effectiveness mainly on human-made objects in indoor scenes, which i) frequently come into contact with us in daily life and are associated with many important applications including perception [1, 2, 3, 4] and manipulation [5, 6, 7], ii) have abundant publicly available 3D models in existing datasets [7, 8, 9], and iii) have been annotated with different types of knowledge according to conventional annotation schemes, which makes it convenient to compare with annotations provided by our scheme. As general concepts are implicit in the fabrication process for human-made objects, the idea of ConceptFactory can adapt well to these object. Contrary to human-made objects, natural objects such as flowers and trees are widely studied in 2D tasks but rarely in 3D tasks. We assume the reasons are that i) the data (*e.g.* scans) of such objects are very difficult to acquire and ii) these objects do not involve in many manipulation tasks, which narrow the applications of these objects. Therefore, there are few 3D models for natural objects in existing publicly available repositories, and currently we do not take the natural objects into consideration when developing ConceptFactory. However, we believe it is possible for the idea of ConceptFactory to cover these objects as they still follow certain regular geometric patterns [10]. We will continue to work on extending ConceptFactory to natural objects to make our idea stronger.

### A.3 Limitations and societal impacts

As mentioned in Sec. A.2, ConceptFactory is currently developed on human-made objects and does not include natural objects. As a future work, we will extend STL-C with more concept templates to cover natural objects, and expand ConceptFactory assets with natural objects by collecting 3D models for such objects and then conceptualizing them.

We hope ConceptFactory can promote machine intelligence to learn comprehensive object knowledge from both vision and robotics aspects, through which we will be brought one step closer to empowering intelligent agents to understand the physical world and therefore improves the life quality for people. However, as the amount of knowledge that machine intelligence needs to learn increases, the consumption of training neural networks will increase accordingly and therefore induce additional economic and energy costs.

### A.4 Compute resources, Licenses and hourly wage for volunteers

For compute resources, our experiments are conducted on a single RTX3090 GPU and Intel(R) Xeon(R) Gold 6133 CPU @ 2.50GHz CPU. The data that used in this study are all publicly available, and are used under their licenses for the current study. There are no personally identifiable information

or offensive content in these data. The hourly wage of volunteers for object conceptualization is about 10 dollars.

# B  Experiments

## B.1  Detailed experiment settings

### B.1.1  Vision Task Settings

**Semantic Segmentation.**  We introduce Point-Transformer [2] and PointNet++ [11] as baselines for semantic segmentation. Point-Transformer proposed a highly expressive neural network design invariant to permutation and cardinality. PointNet++ is an popular and efficient network which servers as the backbone of many 3D frameworks.

In Tab. 1, we give the detailed statistics of certain data in ConceptFactory asset used for this task. The original labels of the objects are acquired by conventional annotation schemes [12, 13, 14]. We obtain our annotations for an object by first assigning each geometry component in a concept template its semantic label, and then propagating such labels to the object's point cloud as mentioned in Sec. 3.4 of the main paper. We uniformly sample 2048 points as the input.

Mean accuracy (mAcc) and mean IoU (mIoU) are adopted as evaluation metrics following the baselines.

|       | Bot | Box | Bkt | Chr | Dsw | Dsp | Dpl | Dor | Egl | Glb | Ket | Ktp | Ltp | Lgt | Mcw |
|-------|-----|-----|-----|-----|-----|-----|-----|-----|-----|-----|-----|-----|-----|-----|-----|
| Train | 217 | 18  | 22  | 425 | 30  | 33  | 193 | 22  | 42  | 25  | 18  | 17  | 33  | 18  | 11  |
| Test  | 94  | 8   | 10  | 183 | 14  | 15  | 83  | 10  | 19  | 12  | 8   | 8   | 15  | 9   | 5   |
| Total | 311 | 26  | 32  | 608 | 44  | 48  | 276 | 32  | 61  | 37  | 26  | 25  | 48  | 27  | 16  |
|       | Mug | Pen | Plr | Rfg | Saf | Scs | Stp | Stf | Swt | Tab | Tcn | USB | Wsm | Win | TTL |
| Train | 83  | 20  | 12  | 14  | 18  | 29  | 15  | 106 | 35  | 37  | 37  | 30  | 11  | 35  | 1606 |
| Test  | 36  | 9   | 6   | 7   | 8   | 13  | 7   | 46  | 15  | 17  | 17  | 14  | 6   | 16  | 710 |
| Total | 119 | 29  | 18  | 21  | 26  | 42  | 22  | 152 | 50  | 54  | 54  | 44  | 17  | 51  | 2316 |

Table 1: Detailed statistics of data used for semantic segmentation task.

**Cross Category Segmentation.**  We adopt GAPartNet [4] as baseline for cross category segmentation. GAPartNet introduced a large-scale part-centric interactive dataset with part-level annotations including poses and cross category semantics, and proposed a robust 3D segmentation method from the perspective of domain generalization by integrating adversarial learning techniques.

The original GAPartNet cross category semantic annotations are obtained by human experts through laboriously data cleaning and manually annotating. In contrast, we can describe fine-grained cross category semantics effortlessly by shared geometric components in concept templates among different categories, and assign cross category semantic labels on our concept template. We then propagate the cross category labels to corresponding point on objects and sample 20000 points with farthest-point-sampling technique as the input.

The detailed statistics of data in ConceptFactory asset used for cross category segmentation is shown in Tab. 2. We follow GAPartNet for data and categories as long as they are available in ConceptFactory asset.

We use widely adopted AP (average precision) as metric for each detectable part. We also report the AP averaged over IoU thresholds from 50% to 95% with a step of 5% for comprehensive evaluation.

**Part Pose Estimation.**  Since GAPartNet [4] also provides a delicately designed part pose estimation mechanism, we still adopt GAPartNet as baseline here. The statistics of data used for part pose estimation is the same to cross category segmentation task, as is shown in Tab. 2.

Similar to the acquisition of the original cross category semantic annotation, the original part pose annotations in GAPartNet are acquired with human labor to clean data and subsequently aligning and annotating part poses. As comparison, our part pose annotations are acquired in the following way. Through object conceptualization, we have obtained the concept template instances for each part of an object. Each template instance has its intrinsic reference coordinate system (*e.g.* the edge of a cuboid should be along with $x, y, z$ axis in its reference coordinate system). We can define the pose

of each detectable part of the object as the transformation from the concept's reference coordinate system to the global coordinate system.

We report metrics including rotation error ($R_e$), translation error ($T_e$), scale error ($S_e$), 3D mIoU, (5°, 5cm) accuracy ($A_5$) and (10°, 10cm) accuracy ($A_{10}$) following GAPartNet.

| Seen Cat | Bkt | Box | Dor | Kpt | Mcw | Rfg | Saf | Stf | Tcn | Wsm | TTL |
|---|---|---|---|---|---|---|---|---|---|---|---|
| Train | 25 | 20 | 7 | 10 | 13 | 28 | 14 | 117 | 26 | 14 | 274 |
| Test | 6 | 5 | 7 | 10 | 3 | 29 | 15 | 29 | 26 | 3 | 133 |
| Total | 31 | 25 | 14 | 20 | 16 | 57 | 29 | 146 | 52 | 17 | 407 |
| Unseen Cat | Dsw | Lpt | Tab | | | | | | | | TTL |
| Test/Total | 8 | 48 | 77 | | | | | | | | 133 |

Table 2: Detailed statistics of data used for GAPartNet related tasks: cross category segmentation, part pose estimation and manipulation tasks with GAPartNet as baseline.

### B.1.2   Manipulation Task Settings

We adopt Where2Act [5], Where2Explore [6], as well as the aforementioned GAPartNet [4] as baselines for manipulation task.

**Where2Act & Where2Explore.**   Where2Act proposed a learning-from-interaction framework with an online data sampling strategy that enables training in simulation and generalizing across categories. Where2Explore further proposed an affordance learning framework that effectively explores novel categories with minimal interactions on instances, transferring affordance knowledge to similar parts of objects across different categories.

The affordance labels of Where2Act and Where2Explore are not provided along with articulated objects. Instead, affordance labels of an object are explored through pixel-wise interaction with the object in a simulated environment. This approach may result in inaccurate and noisy labels due to random sampling and imperfections of the simulator. In comparison, we acquire the affordance labels by first determining the affordable region within the concept template and subsequently propagating this affordance to the object's point cloud as mentioned in Sec. 3.4 of the main paper. We use farthest-point-sampling technique to sample 10000 points for training and evaluation following the setting of baselines.

As shown in Tab. 3, totally 15 representative categories of objects from PartNet-Mobility [14] are used in experiments following the baselines, after removing categories that are typically too small (*e.g.* Pen, USB) or do not make sense for a single-gripper to manipulate (*e.g.* Bottle, Scissors). A full list of the specific manipulation tasks for Where2Act and Where2Explore are provided in Tab. 4, which can be categorized into two general action types: pushing and pulling.

| Train Cats | Box | Dor | Fct | Ket | Mcw | Rfg | Stf | Swt | Tcn | Win | TTL |
|---|---|---|---|---|---|---|---|---|---|---|---|
| Train | 20 | 23 | 65 | 22 | 9 | 32 | 113 | 53 | 52 | 40 | 429 |
| Test | 8 | 12 | 19 | 7 | 3 | 11 | 36 | 17 | 17 | 18 | 148 |
| Total | 28 | 35 | 84 | 29 | 12 | 43 | 149 | 70 | 69 | 58 | 577 |
| Test Cats | Bkt | Kpt | Saf | Tab | Wsm | | | | | | TTL |
| Test/Total | 36 | 23 | 29 | 95 | 16 | | | | | | 199 |

Table 3: Detailed statistics of data used for manipulation tasks with Where2Act and Where2Explore as baselines.

**GAPartNet.**   GAPartNet determines the interaction target and trajectory according to the knowledge of generalizable and actionable parts in objects, namely GAParts.

Here we follow the implementation of data preparation as aforementioned in Sec. B.1.1 since the network and training data for detecting GAParts in manipulation task is the same as cross category segmentation and part pose estimation. The list of specific tasks for manipulation using GAPartNet is provided in Tab. 5.

| Category | Tasks |
| --- | --- |
| Box | Push/Pull Lid |
| Bucket | Push/Pull Handle |
| Door | Push Door; Push/Pull Door via Handle |
| Faucet | Push/Pull Switch |
| Kettle | Push/Pull Handle |
| KitchenPot | Push/Pull Handle; Pull Lid |
| Microwave | Push Door; Push/Pull Door via Handle |
| Refridgerator | Push Door; Push/Pull Door via Handle |
| Safe | Push Door; Push/Pull Door via Handle |
| StorageFurniture | Push Door; Push/Pull Door via Handle; Push/Pull Drawer via Handle |
| Switch | Push/Pull Switch |
| Table | Push Closet; Push/Pull Closet via Handle |
| TrashCan | Push/Pull Lid |
| WashingMachine | Push Door; Push/Pull Door via Handle; Push Lid |
| Window | Push Window; Push/Pull Window via Handle |

Table 4: List of specific manipulations tasks with Where2Act and Where2Explore as baselines. The tasks can be generally categorized into pushing and pulling.

| Category | Tasks |
| --- | --- |
| Box | Rotate Hinge Lid |
| Bucket | Rotate Hinge Handle |
| DishWasher | Close Hinge Door; Open/Close Door via Line Fixed Handle |
| Door | Close Hinge Door; Open/Close Door via Line/Round Fixed Handle |
| Kettle | Rotate Hinge Handle |
| KitchenPot | Move Lid via Line Fixed Handle/Round Fixed Handle |
| Laptop | Rotate Hinge Lid |
| Microwave | Close Hinge Door; Open/Close Hinge Door via Line Fixed Handle |
| Refrigerator | Close Hinge Door; Open/Close Hinge Door via Line Fixed Handle |
| Safe | Close Hinge Door; Open/Close Hinge Door via Hinge Knob |
| StorageFurniture | Close Hinge Door; Open/Close Hinge Door via Line/Round Fixed Handle; Close Slider Drawer; Open/Close Slider Drawer via Line/Round Fixed Handle |
| Table | Close Hinge Door; Open/Close Hinge Door via Line/Round Fixed Handle; Close Slider Drawer; Open/Close Slider Drawer via Line/Round Fixed Handle |
| TrashCan | Rotate Hinge Lid; Rotate Hinge Lid via Line Fixed Handle |
| WashingMachine | Close Hinge Door; |

Table 5: List of specific manipulations tasks with GAPartNet. The tasks are defined based on GAParts.

**Environment & Metric.** We adopt SAPIEN simulator [14] as the interaction environment for manipulation tasks and corresponding environment settings following the baselines. For each round of interaction simulation, we initially place an object in SAPIEN simulator at the center of the scene. The joint state of the object has a 50% chance of being closed (e.g. a closed door) and another 50% chance of being open at a random extent (e.g. a half open drawer). The whole scene is observed through an RGB-D camera with known intrinsic parameters. The camera stares at the center of the object and is located at the upper hemisphere with random azimuth $[0°, 360°)$ and random altitude $[30°, 60°]$. A Franka Panda flying gripper with 2 fingers is used to interact with the object. For pushing tasks, a closed gripper is initially placed 0.05m away from the target along the movement direction, then moves forward for a longer distance in order to push the target. For pulling tasks, an open gripper is placed 0.05m away from the the target along the movement direction, then moves towards the target for 0.045m and closes itself to grasp the target. The gripper subsequently moves back to the start point to pull the target.

Success rate is used as the evaluation metric. Following the baselines, we consider an interaction with a certain part successful if either of the following condition is hold: 1) absolute motion of the part is greater than 0.01. 2) motion of the part is greater than half of its maximum motion range.

**Vision Task Results.**    We provide detailed semantic segmentation results for each object category in Tab. 6. Since cross category segmentation and part pose estimation in GAPartNet are not object category level tasks, we provide results per part in Tab. 7 for cross category segmentation and detailed 6-dof pose related errors in Tab. 8 for part pose estimation. For all tasks, the results show only minor discrepancies between baselines and our approach. This indicates that annotations provided by our scheme possess high quality on par with original ones. We attribute a slight performance decline in some cases to the cognitive variance between our annotators for object conceptualization and the annotators involved in obtaining the original annotations, as we use the original annotations as ground truth to evaluate the performance of both models. For certain performance improvements, we consider the fact that our mathematically grounded annotations can offer better consistency and lower noise, which supervise networks for better convergence.

| PointTransformer | **mAcc**(%) ↑ | | | | | | | | | | | | | |
|---|---|---|---|---|---|---|---|---|---|---|---|---|---|---|
| | Bot | Box | Bkt | Chr | Dsw | Dsp | Dpl | Dor | Egl | Glb | Ket | Ktp | Ltp | Lgt | Mcw |
| Original | 97.3 | 93.3 | 96.7 | 94.0 | 89.9 | 93.4 | 95.7 | 75.8 | 96.7 | 95.4 | 92.3 | 95.0 | 94.3 | 84.4 | 78.8 |
| ConFac | 96.4 | 92.1 | 96.7 | 92.0 | 89.7 | 94.2 | 95.2 | 76.1 | 95.6 | 95.7 | 92.5 | 94.1 | 95.1 | 86.0 | 80.4 |
| Δ | -0.9 | -1.2 | 0.0 | -2.0 | -0.2 | 0.8 | -0.5 | 0.3 | -1.1 | 0.3 | 0.2 | -0.9 | 0.8 | 1.6 | 1.6 |
| | Mug | Pen | Plr | Rfg | Saf | Scs | Stp | Stf | Swt | Tab | Tcn | USB | Wsm | Win | AVG |
| Original | 93.4 | 91.0 | 83.4 | 88.7 | 92.1 | 91.5 | 79.5 | 92.8 | 90.6 | 87.1 | 92.1 | 79.3 | 90.0 | 84.6 | 90.0 |
| ConFac | 94.4 | 90.4 | 81.5 | 89.8 | 93.0 | 91.5 | 79.9 | 91.0 | 89.4 | 87.7 | 92.7 | 77.8 | 89.0 | 84.3 | 89.8 |
| Δ | 1.0 | -0.6 | -1.9 | 1.1 | 0.9 | 0.0 | 0.4 | -1.8 | -1.2 | 0.6 | 0.6 | -1.5 | -1.0 | -0.3 | -0.2 |

| PointTransformer | **mIoU**(%) ↑ | | | | | | | | | | | | | |
|---|---|---|---|---|---|---|---|---|---|---|---|---|---|---|
| | Bot | Box | Bkt | Chr | Dsw | Dsp | Dpl | Dor | Egl | Glb | Ket | Ktp | Ltp | Lgt | Mcw |
| Original | 71.4 | 76.8 | 49.9 | 88.8 | 94.5 | 80.2 | 77.2 | 26.9 | 98.1 | 75.2 | 83.9 | 73.2 | 87.8 | 47.7 | 59.2 |
| ConFac | 71.1 | 75.0 | 49.5 | 87.6 | 93.9 | 82.0 | 76.3 | 28.9 | 97.2 | 77.2 | 84.0 | 72.3 | 89.7 | 49.5 | 62.3 |
| Δ | -0.3 | -1.8 | -0.4 | -1.2 | -0.6 | 1.8 | -0.9 | 2.0 | -0.9 | 2.0 | 0.1 | -0.9 | 1.9 | 1.8 | 3.1 |
| | Mug | Pen | Plr | Rfg | Saf | Scs | Stp | Stf | Swt | Tab | Tcn | USB | Wsm | Win | AVG |
| Original | 98.3 | 78.1 | 72.9 | 92.6 | 65.0 | 95.9 | 69.6 | 92.3 | 91.5 | 71.8 | 75.7 | 79.6 | 52.2 | 82.6 | 75.4 |
| ConFac | 98.2 | 77.4 | 71.5 | 94.5 | 65.7 | 95.7 | 71.3 | 91.5 | 68.3 | 73.5 | 77.1 | 78.6 | 50.6 | 81.5 | 75.6 |
| Δ | -0.1 | -0.7 | -1.4 | 1.9 | 0.7 | -0.2 | 1.7 | -0.8 | -1.3 | 1.7 | 1.4 | -1.0 | -1.6 | -1.1 | 0.2 |

| Pointnet++ | **mAcc**(%) ↑ | | | | | | | | | | | | | |
|---|---|---|---|---|---|---|---|---|---|---|---|---|---|---|
| | Bot | Box | Bkt | Chr | Dsw | Dsp | Dpl | Dor | Egl | Glb | Ket | Ktp | Ltp | Lgt | Mcw |
| Original | 95.9 | 94.9 | 96.9 | 94.0 | 90.1 | 94.2 | 96.0 | 72.5 | 95.0 | 92.4 | 83.0 | 92.2 | 95.0 | 84.4 | 85.0 |
| Confac | 95.3 | 95.7 | 96.8 | 92.5 | 89.7 | 94.6 | 95.6 | 73.0 | 94.2 | 92.7 | 84.3 | 91.6 | 95.0 | 84.4 | 85.9 |
| Δ | -0.6 | 0.8 | -0.1 | -1.5 | -0.4 | 0.4 | -0.4 | 0.5 | -0.8 | 0.3 | 1.3 | -0.6 | 0.0 | 0.0 | 0.9 |
| | Mug | Pen | Plr | Rfg | Saf | Scs | Stp | Stf | Swt | Tab | Tcn | USB | Wsm | Win | AVG |
| Original | 93.2 | 87.9 | 84.2 | 90.8 | 86.3 | 91.0 | 78.9 | 88.8 | 89.0 | 81.4 | 92.5 | 79.3 | 92.0 | 79.6 | 88.8 |
| ConFac | 92.9 | 87.5 | 82.3 | 90.9 | 88.2 | 91.2 | 80.0 | 88.2 | 88.4 | 83.0 | 92.7 | 77.7 | 92.5 | 78.5 | 88.8 |
| Δ | -0.3 | -0.4 | -1.9 | 0.1 | 1.9 | 0.2 | 1.1 | -0.6 | -0.6 | 1.6 | 0.2 | -1.6 | 0.5 | -1.1 | 0.0 |

| Pointnet++ | **mIoU**(%) ↑ | | | | | | | | | | | | | |
|---|---|---|---|---|---|---|---|---|---|---|---|---|---|---|
| | Bot | Box | Bkt | Chr | Dsw | Dsp | Dpl | Dor | Egl | Glb | Ket | Ktp | Ltp | Lgt | Mcw |
| Original | 71.3 | 87.0 | 48.4 | 89.5 | 81.0 | 76.0 | 75.3 | 39.1 | 88.4 | 82.3 | 46.2 | 82.1 | 70.2 | 52.6 | 60.6 |
| ConFac | 70.2 | 88.5 | 48.4 | 88.1 | 80.4 | 76.4 | 74.7 | 37.6 | 87.7 | 82.3 | 47.8 | 81.2 | 70.0 | 52.4 | 61.0 |
| Δ | -1.1 | 1.5 | 0.0 | -1.4 | -0.6 | 0.4 | -0.6 | -1.5 | -0.7 | 0.0 | 1.6 | -0.9 | -0.2 | -0.2 | 0.4 |
| | Mug | Pen | Plr | Rfg | Saf | Scs | Stp | Stf | Swt | Tab | Tcn | USB | Wsm | Win | AVG |
| Original | 89.0 | 67.9 | 74.1 | 63.9 | 53.8 | 58.1 | 59.1 | 75.3 | 46.1 | 67.7 | 65.2 | 71.7 | 42.5 | 65.7 | 67.2 |
| ConFac | 88.9 | 67.7 | 74.0 | 64.0 | 54.9 | 60.0 | 59.6 | 75.0 | 45.5 | 69.2 | 67.2 | 70.9 | 44.3 | 64.2 | 67.3 |
| Δ | -0.1 | -0.2 | -0.1 | 0.1 | 1.1 | 1.9 | 0.5 | -0.3 | -0.6 | 1.5 | 2.0 | -0.8 | 1.8 | -1.5 | 0.1 |

Table 6: Experiment results of semantic segmentation in detail. Δ denotes the gap of between two baseline models separately trained with the original or our annotations in absolute value.

| | Ln.F.HD | Rd.F.HD | Hg.Ld | Sd.Ld | Sd.Dw | Hg.Dr | Avg AP | Avg AP50 |
|---|---|---|---|---|---|---|---|---|
| Original | 69.4 | 36.0 | 39.0 | 68.2 | 51.2 | 84.5 | 30.0 | 37.7 |
| ConFac | 69.6 | 36.4 | 39.0 | 68.7 | 51.9 | 84.9 | 30.6 | 38.5 |
| Δ | 0.2 | 0.4 | 0.0 | 0.5 | 0.7 | 0.4 | 0.6 | 0.8 |

Table 7: Experiment results of cross category segmentation in detail. Δ denotes the gap of between two baseline models separately trained with the original or our annotations in absolute average precision.

**Manipulation Task Results.**    We provide manipulation results with metric sample success rate (ssr) for each object category in detail in Tab. 9 and Tab. 10. The results of Where2Act and Where2Explore highlight considerable improvements by training with our affordance annotations, demonstrating the

|  | $\mathbf{Re}(°)\downarrow$ | $\mathbf{Te}$(cm)$\downarrow$ | $\mathbf{Se}$(cm)$\downarrow$ | $\mathbf{mIoU}(\%)\uparrow$ | $A_5(\%)\uparrow$ | $A_{10}(\%)\uparrow$ |
|---|---|---|---|---|---|---|
| Original | 9.62 | 0.024 | 0.066 | 42.6 | 29.1 | 55.7 |
| ConFac | 9.59 | 0.022 | 0.063 | 42.8 | 30.3 | 56.9 |
| $\Delta$ | 0.03 | 0.002 | 0.003 | 0.2 | 1.2 | 1.2 |

Table 8: Experiment results of part pose estimation in detail. $\Delta$ denotes the gap of between two baseline models separately trained with the original or our annotations in absolute value.

strong ability of our approach to provide accurate, consistent and clean affordance knowledge among various object categories.

| | | Where2Act ssr(%) ↑ | | | | | | | | | | | | | | |
|---|---|---|---|---|---|---|---|---|---|---|---|---|---|---|---|---|
| | | Box | Bkt | Dor | Fct | Ket | Ktp | Mcw | Rfg | Saf | Stf | Swt | Tab | Tcn | Wsm | Win | AVG |
| push | Original | 10.6 | 18.1 | 17.9 | 33.3 | 27.2 | 17.1 | 18.3 | 16.0 | 25.3 | 26.0 | 12.5 | 31.2 | 11.9 | 13.0 | 28.1 | 23.7 |
| | ConFac | 15.2 | 17.3 | 23.4 | 36.8 | 28.1 | 18.8 | 21.9 | 20.0 | 31.5 | 32.9 | 11.5 | 44.0 | 14.5 | 16.4 | 37.5 | 30.0 |
| | $\Delta$ | 4.6 | -0.8 | 5.5 | 3.5 | 0.9 | 1.7 | 3.6 | 4.0 | 6.2 | 6.9 | -1.0 | 12.8 | 2.6 | 3.4 | 9.4 | 6.3 |
| pull | Original | 7.1 | 4.2 | 11.1 | 14.7 | 8.0 | 7.2 | 3.3 | 2.4 | 5.3 | 10.4 | 3.6 | 12.3 | 8.0 | 4.0 | 1.2 | 8.4 |
| | ConFac | 8.3 | 5.2 | 16.4 | 16.2 | 10.1 | 7.3 | 4.2 | 1.7 | 6.7 | 13.5 | 3.3 | 16.1 | 8.6 | 6.7 | 1.0 | 10.6 |
| | $\Delta$ | 1.2 | 1.0 | 5.3 | 1.5 | 2.1 | 0.1 | 0.9 | -0.7 | 1.4 | 3.1 | -0.3 | 3.8 | 0.6 | 2.7 | -0.2 | 2.2 |
| | | Where2Explore ssr(%) ↑ | | | | | | | | | | | | | | |
| | | Box | Bkt | Dor | Fct | Ket | Ktp | Mcw | Rfg | Saf | Stf | Swt | Tab | Tcn | Wsm | Win | AVG |
| push | Original | 16.5 | 26.4 | 35.2 | 23.4 | 16.5 | 25.3 | 51.1 | 27.7 | 30.5 | 51.1 | 11.0 | 41.6 | 21.0 | 8.3 | 15.2 | 33.0 |
| | ConFac | 21.2 | 29.3 | 39.1 | 22.5 | 17.5 | 25.3 | 50.9 | 29.9 | 36.4 | 59.7 | 10.3 | 48.1 | 22.2 | 10.4 | 18.0 | 37.4 |
| | $\Delta$ | 4.7 | 2.9 | 3.9 | -0.9 | 1.0 | 0.0 | -0.2 | 2.2 | 5.9 | 8.6 | -0.7 | 6.5 | 1.2 | 2.1 | 2.8 | 4.4 |
| pull | Original | 16.5 | 17.1 | 11.7 | 16.2 | 17.4 | 14.3 | 11.8 | 9.6 | 9.0 | 13.6 | 2.2 | 25.6 | 19.7 | 7.2 | 1.5 | 15.6 |
| | ConFac | 18.4 | 18.4 | 15.1 | 15.3 | 20.4 | 18.8 | 12.6 | 8.8 | 10.0 | 19.7 | 2.0 | 28.6 | 26.1 | 10.0 | 1.4 | 18.5 |
| | $\Delta$ | 1.9 | 1.3 | 3.4 | -0.9 | 3.0 | 4.5 | 0.8 | -0.8 | 1.0 | 6.1 | -0.2 | 3.0 | 6.4 | 2.8 | -0.1 | 2.9 |

Table 9: Experimental results on manipulation tasks with Where2act and Where2Explore in detail. $\Delta$ denotes the gap of between two baseline models separately trained with the original or our annotations in absolute ssr.

| | Box | Bkt | Dor | Dsw | Ktp | Ltp | Mcw | Rfg | Saf | Stf | Tab | Tcn | Wsm | AVG |
|---|---|---|---|---|---|---|---|---|---|---|---|---|---|---|
| Original | 21.7 | 33.8 | 27.9 | 27.1 | 25.1 | 15.1 | 32.4 | 29.5 | 29.9 | 28.2 | 16.0 | 30.9 | 40.0 | 24.8 |
| ConFac | 25.1 | 36.8 | 27.0 | 27.1 | 25.9 | 17.2 | 34.0 | 30.0 | 29.2 | 29.1 | 15.0 | 30.8 | 39.4 | 25.1 |
| $\Delta$ | 3.4 | 3.0 | -0.9 | 0.0 | 0.8 | 2.1 | 1.6 | 0.5 | -0.7 | 0.9 | -1.0 | -0.1 | -0.6 | 0.3 |

Table 10: Experimental results on manipulation tasks with GAPartNet in detail. We do not distinguish pushing and pulling here following the settings of GAPartNet. $\Delta$ denotes the gap of between two baseline models separately trained with the original or our annotations in absolute ssr.

**Error bars** We provide error bars for our experiments in the main context in Tab. 11 and Tab. 12.

| | Semantic Segmentation | | | | Cross Category Segmentation | | Part Pose Estimation | | |
|---|---|---|---|---|---|---|---|---|---|
| | PointTransformer | | PointNet++ | | GAPartNet | | | | |
| | mAcc | mIoU | mAcc | mIoU | mAP | mAP$_{50}$ | mIoU | $A_5$ | $A_{10}$ |
| Original | ±0.11 | ±0.10 | ±0.08 | ±0.05 | ±0.04 | ±0.08 | ±0.04 | ±0.05 | ±0.06 |
| ConFac | ±0.08 | ±0.12 | ±0.08 | ±0.06 | ±0.03 | ±0.10 | ±0.07 | ±0.05 | ±0.05 |

Table 11: Error bar of experiments results of semantic segmentation, cross category segmentation and part pose estimation.

| | Where2Act | Where2Explore | GAPartNet |
|---|---|---|---|
| Original | ±0.42 / ±0.50 | ±0.36 / ±0.45 | ±0.42 |
| ConFac | ±0.36 / ±0.42 | ±0.33 / ±0.38 | ±0.39 |

Table 12: Error bar of experimental results of manipulation tasks.

## B.3 Annotation visualizations of different knowledge types across different object categories

We provide annotation visualizations of different knowledge types across different object categories in Fig. 1 and Fig. 2, where each annotation pair consist of original annotation (Left) and annotation obtained by our approach (Right). The annotations are divided into three piles according to their types: semantic segmentation, part pose and affordance.

**Semantic Segmentation.** We use different colors to distinguish the semantic regions according to the annotations. We can observe that our approach is able to provide annotation that is almost identical to the original one in most cases. For those annotations with certain gaps such as *chairs* in Row 2, Col 1-2, the original annotations confuse with the definition of crossbars between legs while our approach reasonably and consistently label these regions to part of legs.

**Part Pose.** We use oriented bounding boxes drawn in red lines to represent the pose of each part. The annotation pairs show that our approach is able to assign proper poses to each part.

**Affordance.** Here red regions represent the affordance of different object categories. The conventional method [5] to obtain affordance is based on pixel-level interactions with an object within a simulated environment. However, the randomness of sampling and the imperfections inherent in the simulator can result in affordance labels that are filled with noise. Compared with original annotations, our approach provides more consistent affordance labels with much less noise. This facilitates baseline models in better learning object affordance knowledge and thereby achieving stronger manipulation capabilities.

### B.4 Human evaluation

We further provide a human evaluation to compare the quality of the original and our annotations. We invite five volunteers, show pairs of annotations to them, and ask them to evaluate which is better in each pair. The number of pairs of semantic segmentation, part pose and affordance annotations are 500, 200, and 200 respectively. The results are provided in Tab. 13. For semantic segmentation and part pose, the annotation quality is comparable between the original ones and ours. For affordance knowledge, the annotations acquired by our approach greatly surpass the quality of the original ones. Therefore, we regard this capability of providing affordance knowledge as the major indication of the advantages of our annotation scheme.

|  | Equally Better | Ours Better | Original Better |
|---|---|---|---|
| Semantic Segmentation | 73.6% | 13.4% | 13.0% |
| Part Pose | 91.3% | 4.8% | 3.9% |
| Affordance | 23.3% | 75.0% | 1.7% |

Table 13: Human evaluation of the annotation quality.

### B.5 User Studies for Conceptualization Platform

In order to consistently improve the user experience of our conceptualization platform, we have invited a total of 23 volunteers as test users and conducted two rounds of overall improvements on our platform. Specifically, during each round of improvement, we ask the test users to conceptualize sample objects across all available categories using the platform, and provide feedback such as difficulties during usage, bug reports, as well as suggestions. We then collect the feedback and use it for improving our platform. To verify the effectiveness of our improvements, we additionally invite the test users to fill out System Usability Scales [15] and calculate the average SUS score before and after each round of improvement. An illustration of our specific changes on the platform as well as the SUS scores are shown in Fig.3.

## C STL-C and ConceptFactory asset

Tab. 14 is the cross reference table explaining the notations in Tab. 1 of the main paper. In Fig. 4-16, we show for all the 39 object categories that i) how an object category is divided into a group of parts according to their structural for better guidance in the conceptualization process (blue solid line), ii) how an object category can be described and covered by a series of concept templates (black solid line), and iii) how concept templates are organized by geometry components (black dashed line).

# Semantic Segmentation

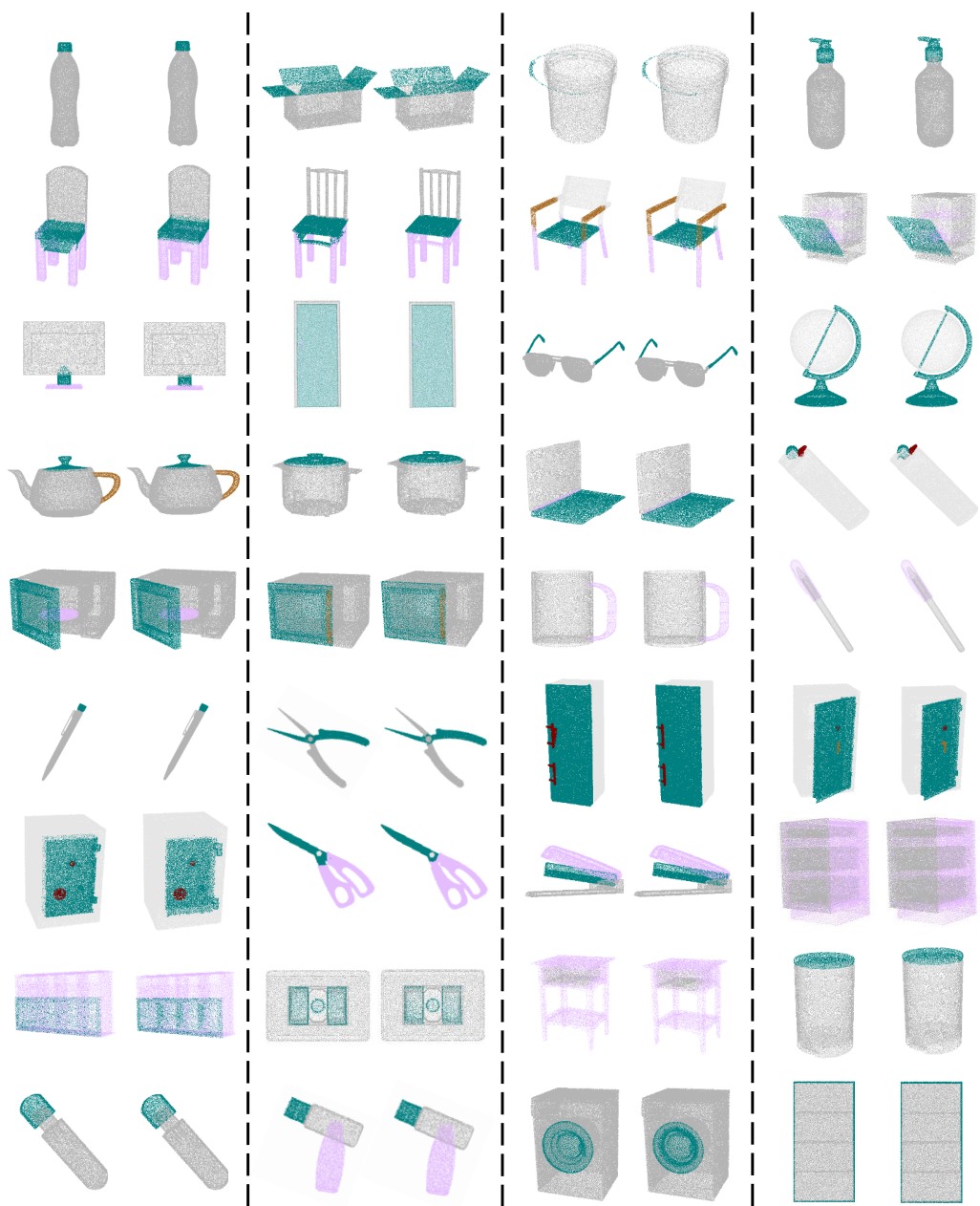

Figure 1: Annotation visualization Part I: Semantic Segmentation.

**Part Pose**

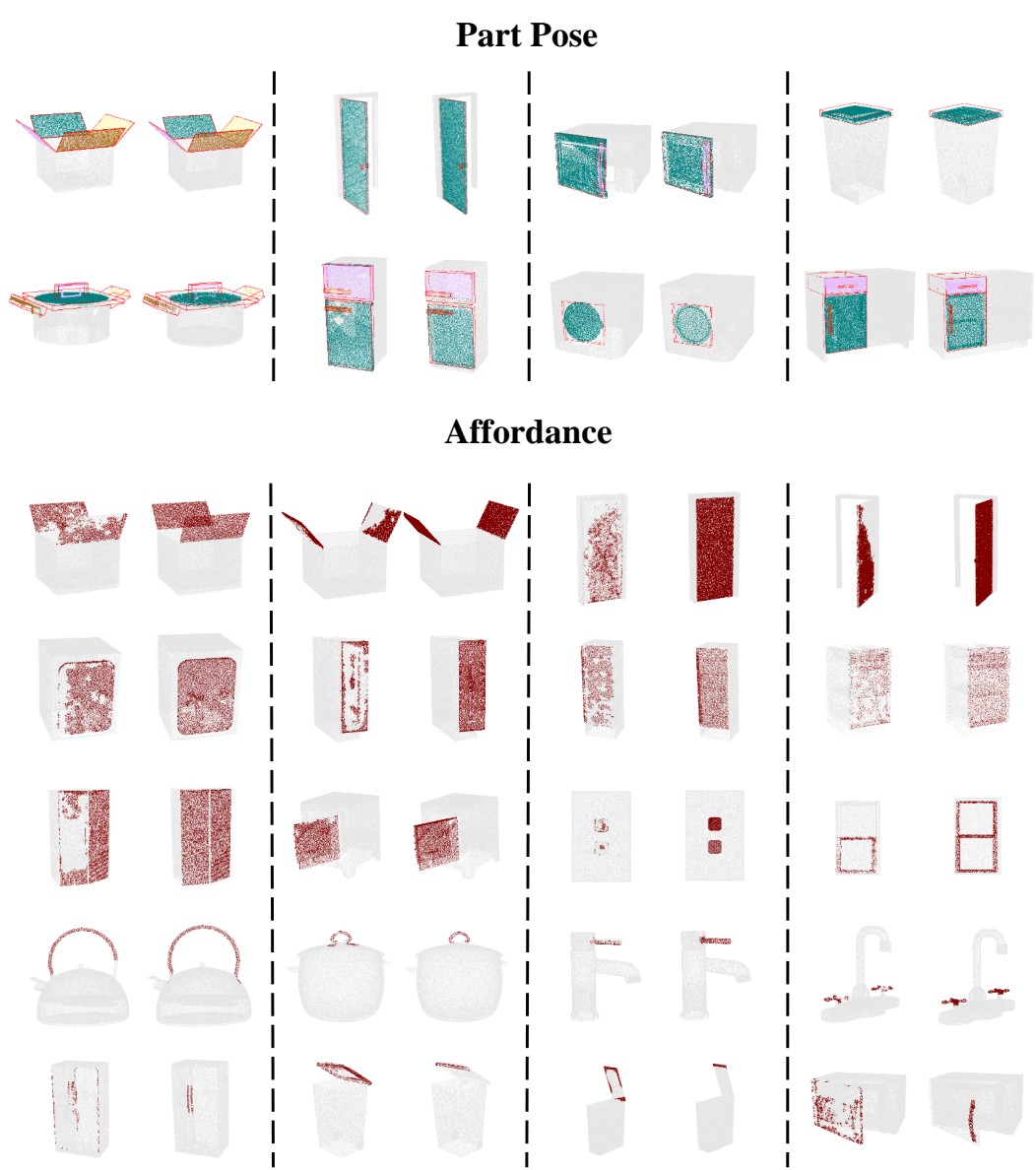

**Affordance**

Figure 2: Annotation visualization Part II: Part Pose and Affordance (Push - Row.1-3, Pull - Row.4-5).

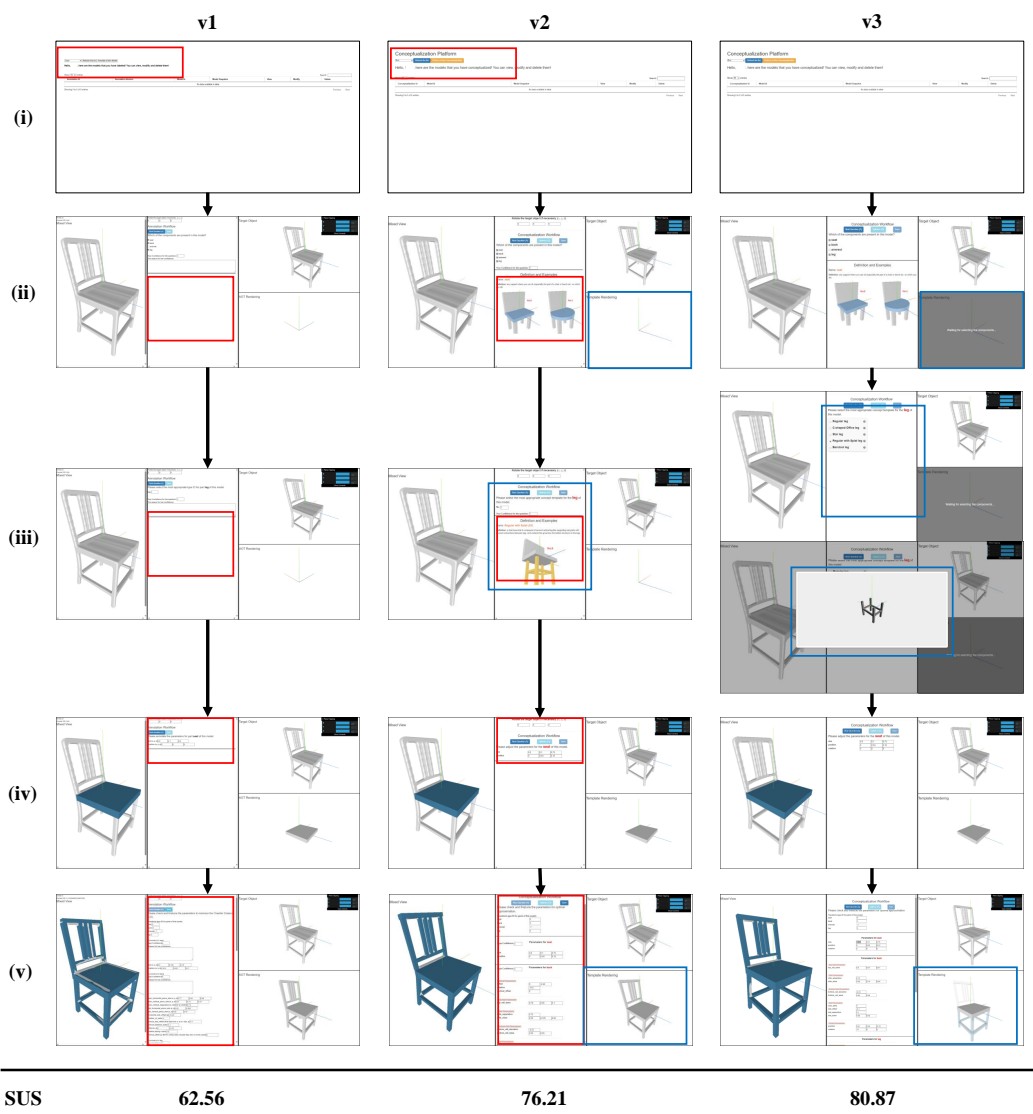

Figure 3: **[Top]** Page-wise (i-v) illustrations of our platform's evolvement over two rounds of improvements. Key differences between versions are highlighted using boxes of the same color. **[Bottom]** The average SUS scores of the three versions.

| Notation | Full name | Notation | Full name | Notation | Full name |
|---|---|---|---|---|---|
| Bot | Bottle | Box | Box | Bkt | Bucket |
| Chr | Chair | Clp | Clip | Dsw | DishWasher |
| Dsp | Dispenser | Dpl | Display | Dor | Door |
| Drh | DoorHandle | Egl | Eyeglasses | Fct | Faucet |
| Fdr | FoldingRack | Glb | Globe | Gsk | GlueStick |
| Ket | Kettle | Ktp | KitchenPot | Knf | Knife |
| Ltp | Laptop | Lgt | Lighter | Mcw | Microwave |
| Mug | Mug | Ovn | Oven | Pen | Pen |
| Plr | Pliers | Rfg | Refrigerator | Rlr | Ruler |
| Saf | Safe | Scs | Scissors | Smp | Shampoo |
| Shv | Shaver | Stp | Stapler | Stf | StorageFurniture |
| Swt | Switch | Tab | Table | Tcn | Trashcan |
| USB | USB | Wsm | WashingMachine | Win | Window |

Table 14: Cross reference table of notations in Tab. 1 of the main paper.

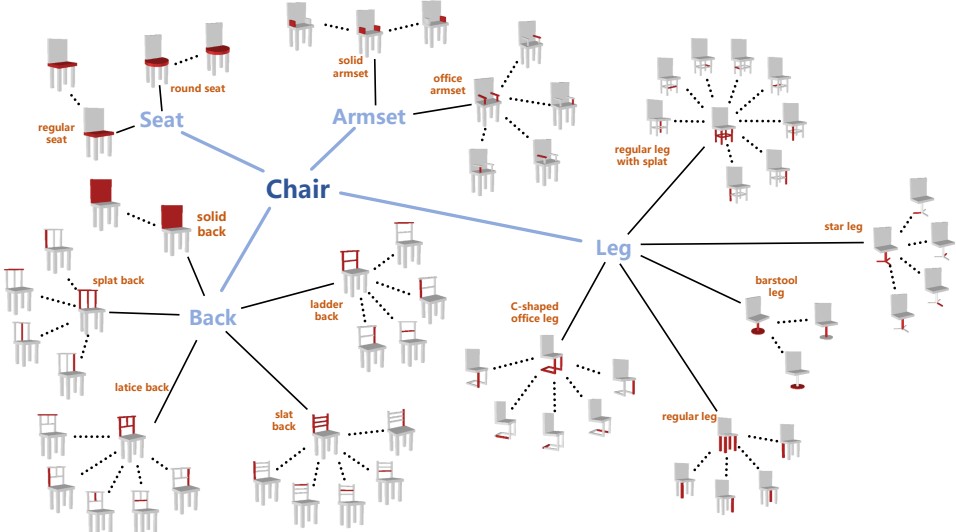

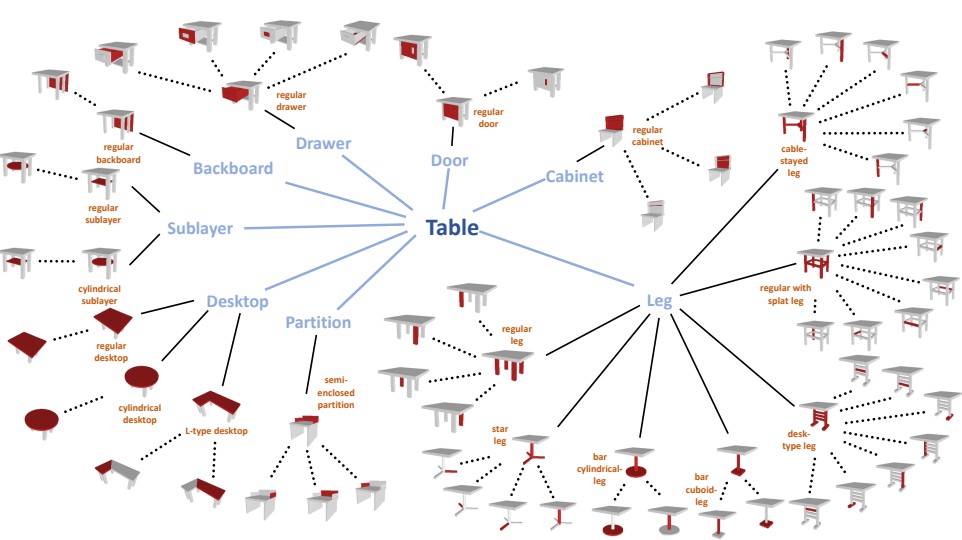

Figure 4: Visualizations for **Chair and Table** about i) how they are divided into a group of parts according to their structural for better guidance in the conceptualization process (blue solid line), ii) how they can be described and covered by a series of concept templates (black solid line), and iii) how concept templates are organized by geometry components (black dashed line).

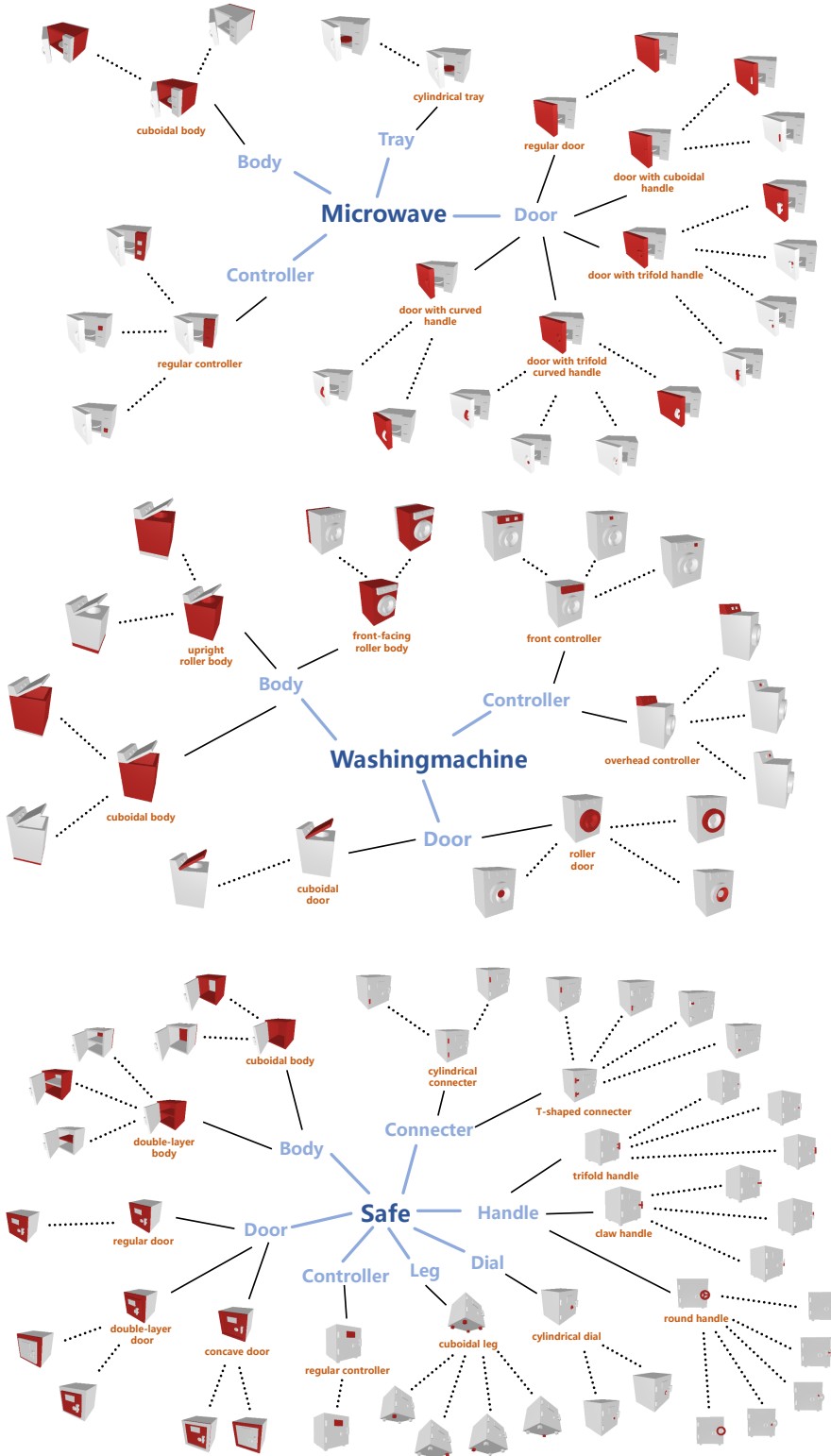

Figure 5: Visualizations for **Microwave, WashingMachine and Safe** about i) how they are divided into a group of parts according to their structural for better guidance in the conceptualization process (blue solid line), ii) how they can be described and covered by a series of concept templates (black solid line), and iii) how concept templates are organized by geometry components (black dashed line).

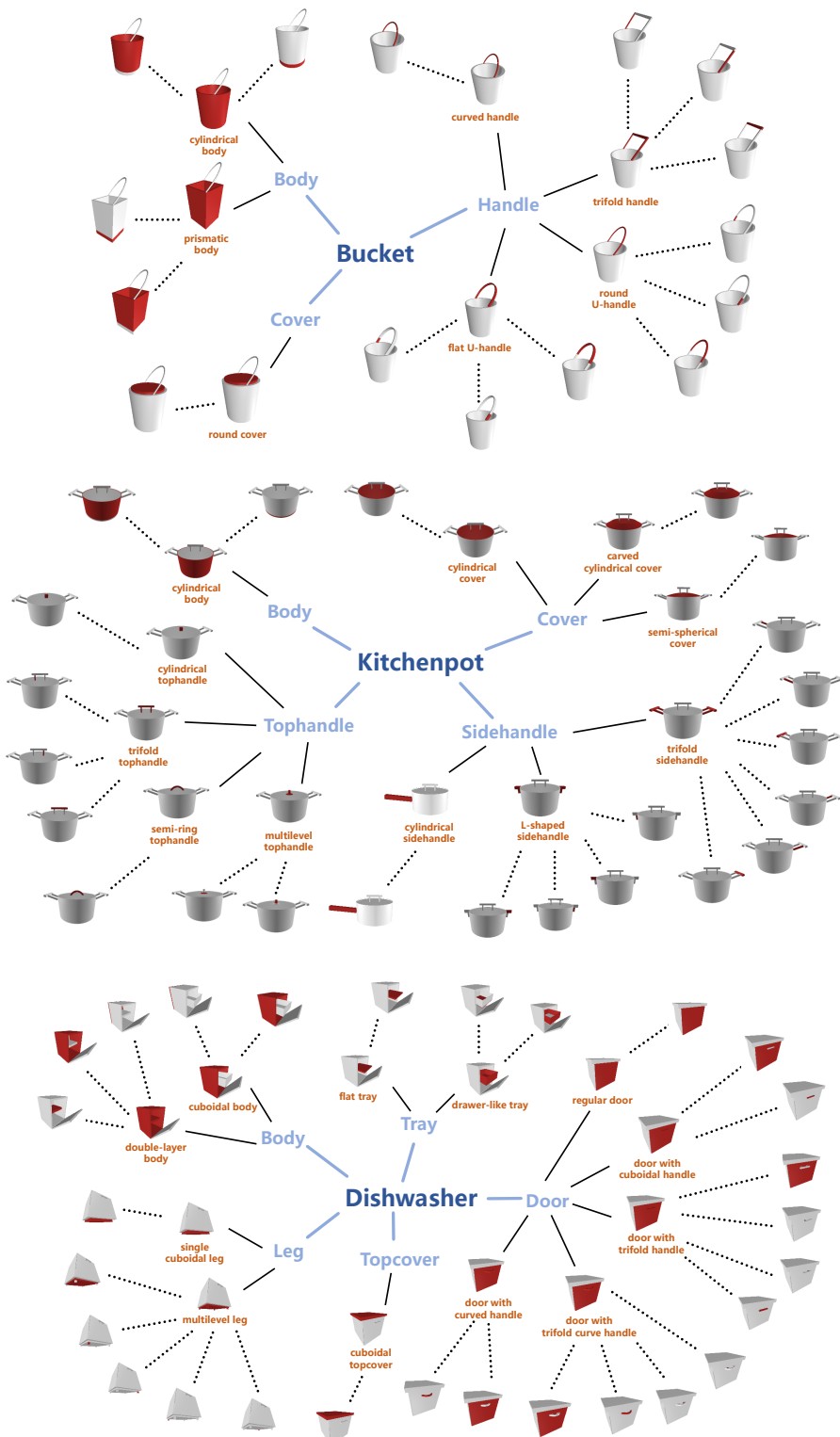

Figure 6: Visualizations for **Bucket, KitchenPot and DishWasher** about i) how they are divided into a group of parts according to their structural for better guidance in the conceptualization process (blue solid line), ii) how they can be described and covered by a series of concept templates (black solid line), and iii) how concept templates are organized by geometry components (black dashed line).

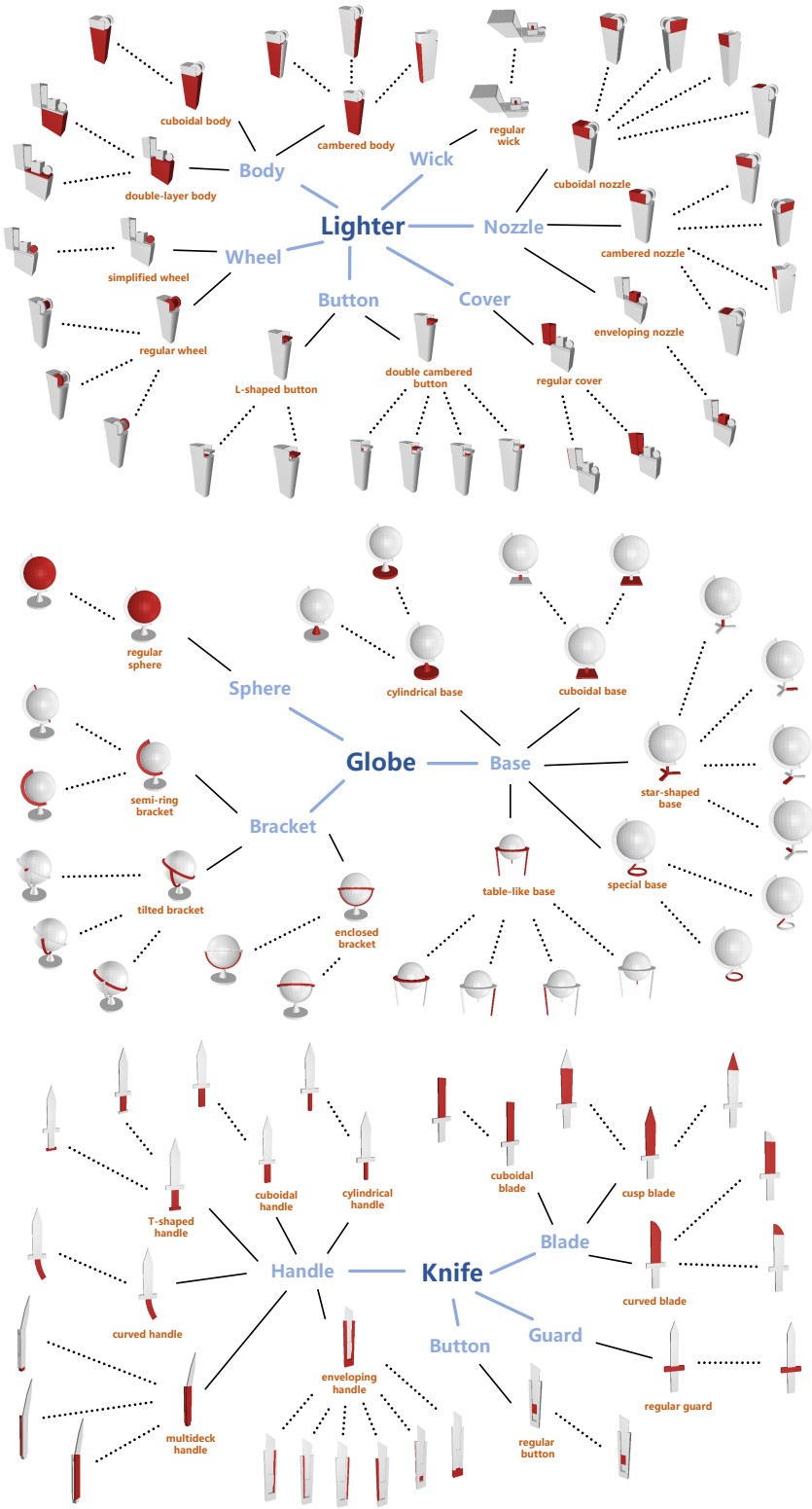

Figure 7: Visualizations for **Lighter, Globe and Knife** about i) how they are divided into a group of parts according to their structural for better guidance in the conceptualization process (blue solid line), ii) how they can be described and covered by a series of concept templates (black solid line), and iii) how concept templates are organized by geometry components (black dashed line).

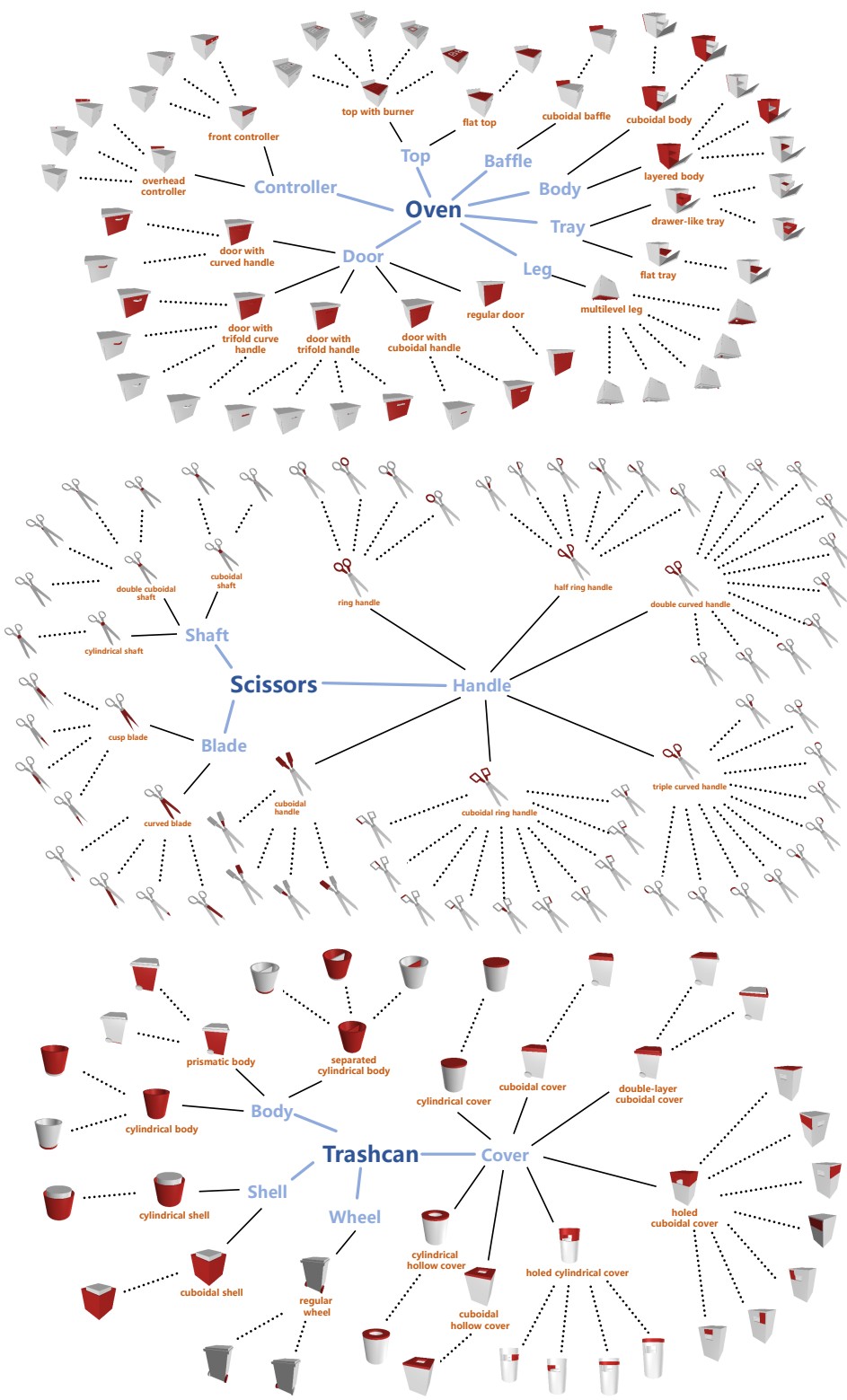

Figure 8: Visualizations for **Oven, Scissors and Trashcan** about i) how they are divided into a group of parts according to their structural for better guidance in the conceptualization process (blue solid line), ii) how they can be described and covered by a series of concept templates (black solid line), and iii) how concept templates are organized by geometry components (black dashed line).

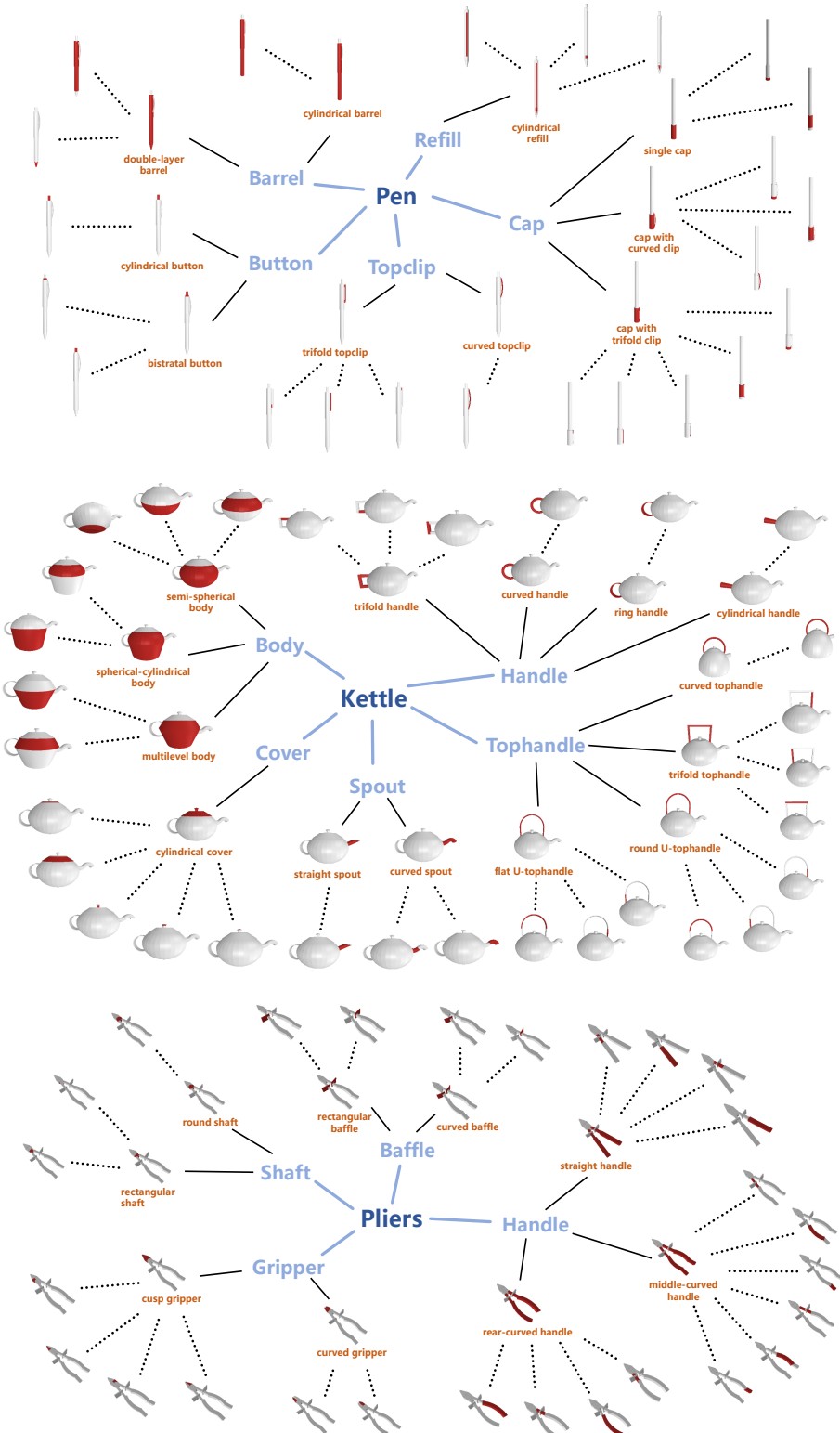

Figure 9: Visualizations for **Pen, Kettle and Pliers** about i) how they are divided into a group of parts according to their structural for better guidance in the conceptualization process (blue solid line), ii) how they can be described and covered by a series of concept templates (black solid line), and iii) how concept templates are organized by geometry components (black dashed line).

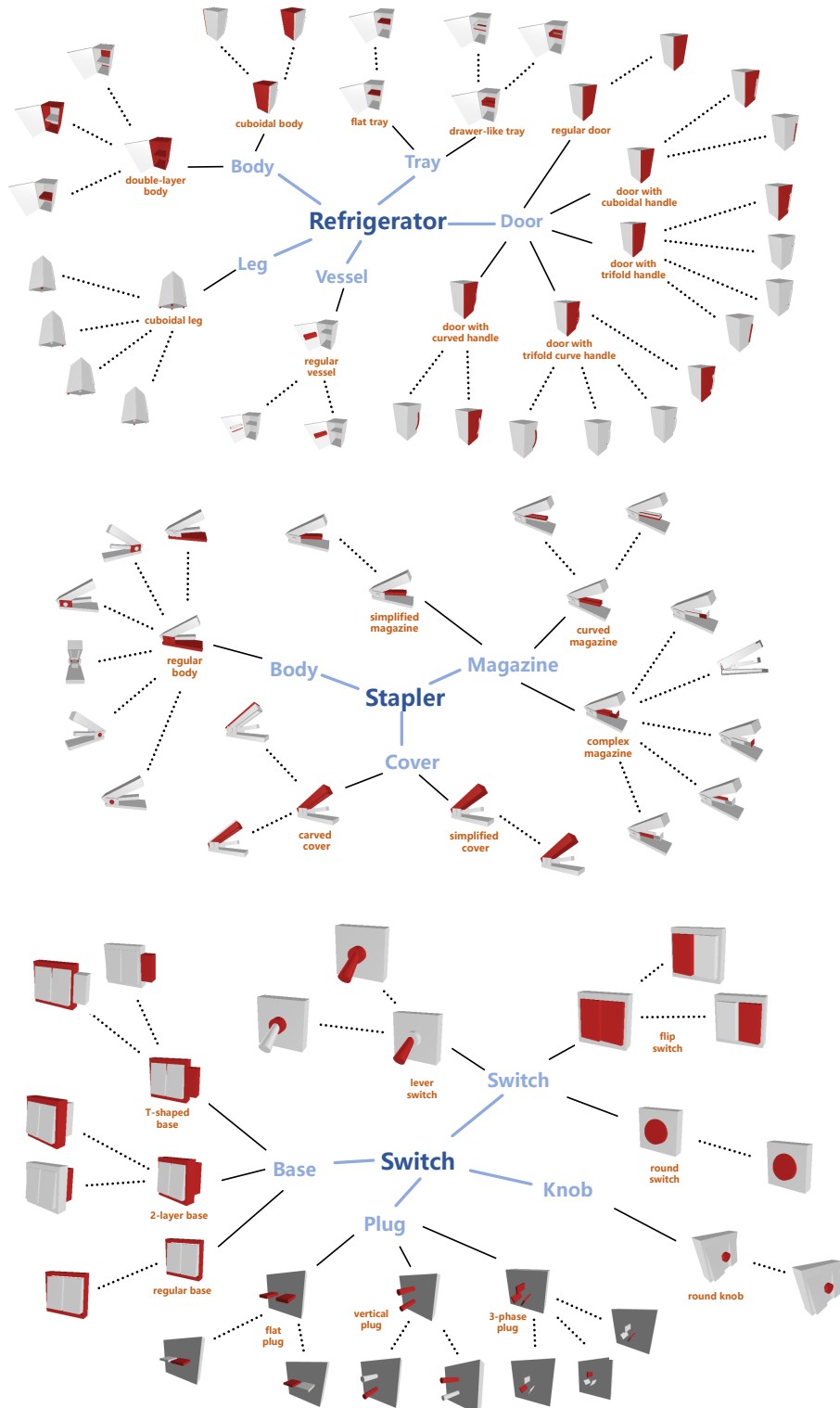

Figure 10: Visualizations for **Refrigerator, Stapler and Switch** about i) how they are divided into a group of parts according to their structural for better guidance in the conceptualization process (blue solid line), ii) how they can be described and covered by a series of concept templates (black solid line), and iii) how concept templates are organized by geometry components (black dashed line).

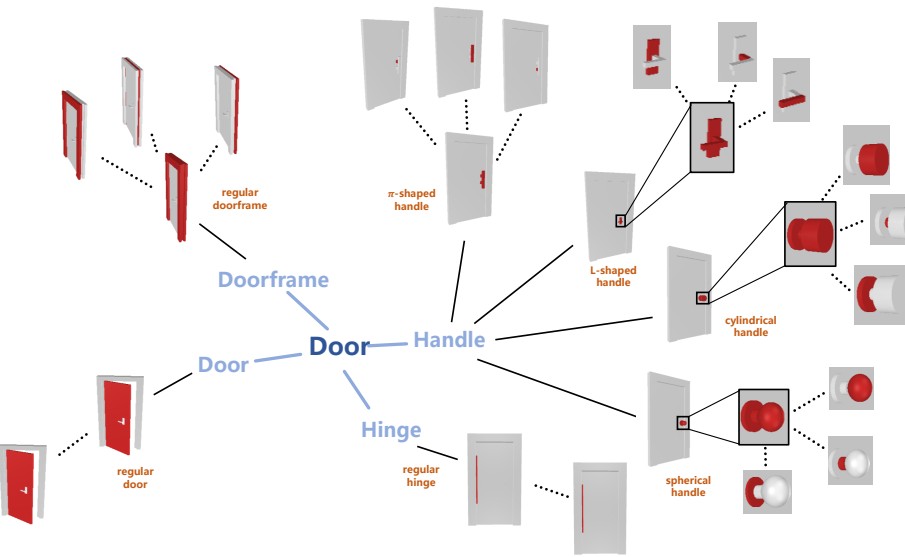

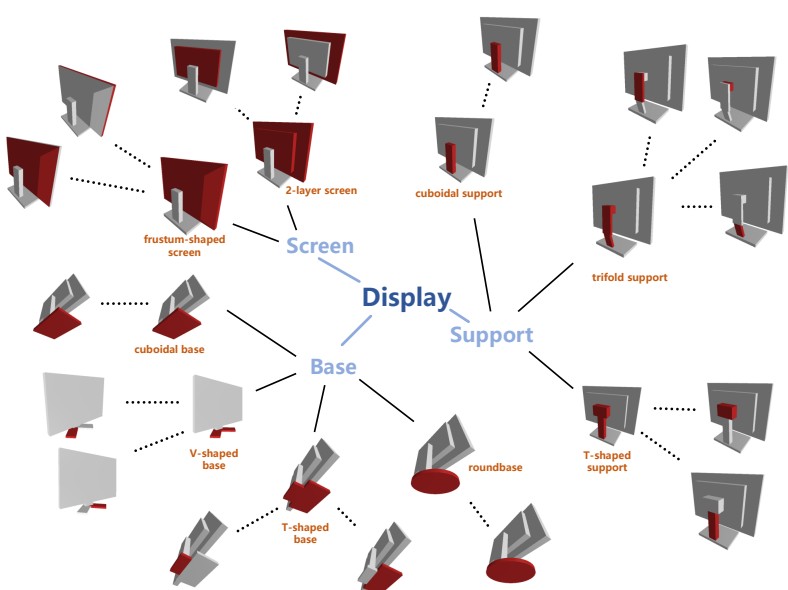

Figure 11: Visualizations for **Door and Display** about i) how they are divided into a group of parts according to their structural for better guidance in the conceptualization process (blue solid line), ii) how they can be described and covered by a series of concept templates (black solid line), and iii) how concept templates are organized by geometry components (black dashed line).

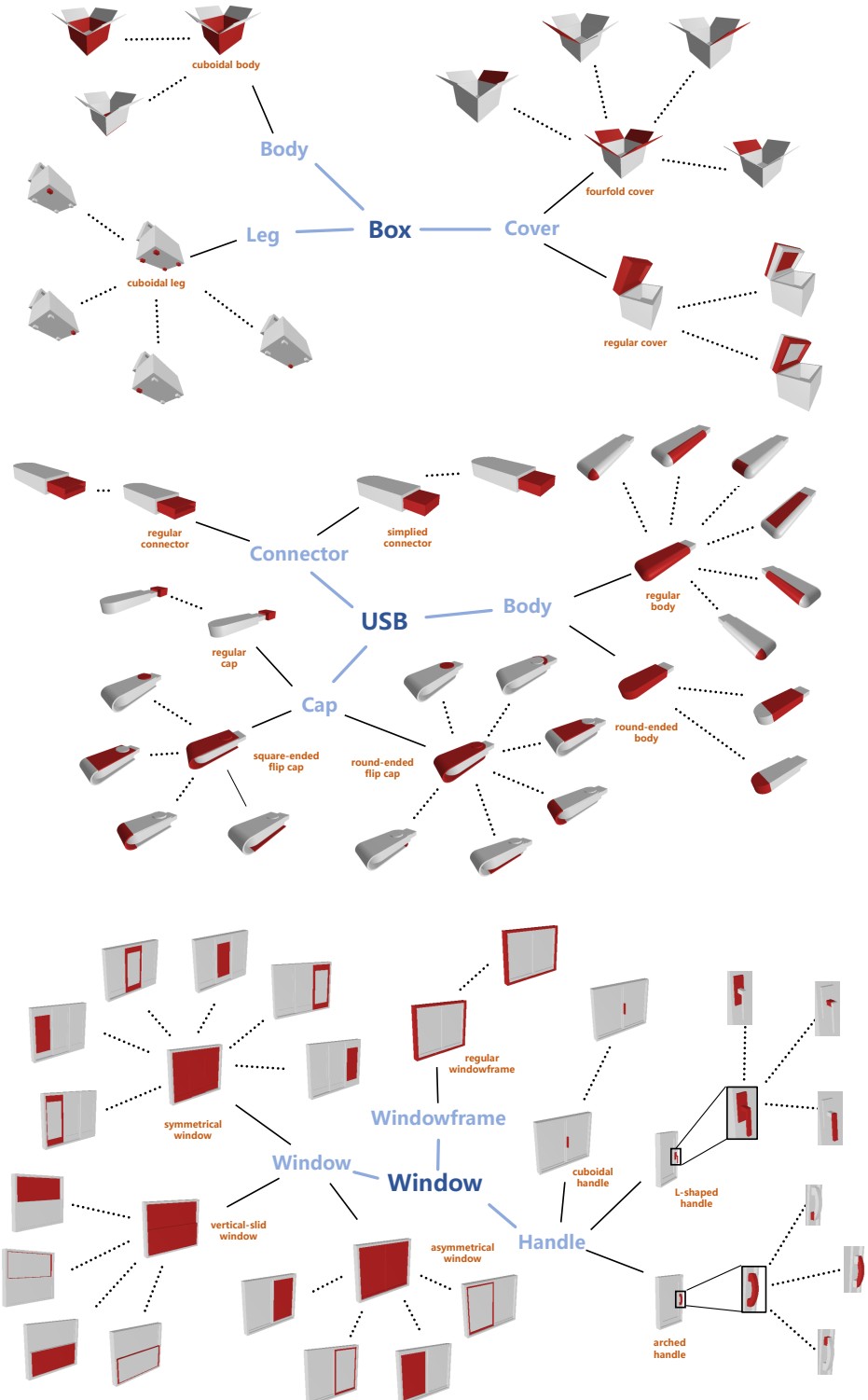

Figure 12: Visualizations for **Box, USB and Window** about i) how they are divided into a group of parts according to their structural for better guidance in the conceptualization process (blue solid line), ii) how they can be described and covered by a series of concept templates (black solid line), and iii) how concept templates are organized by geometry components (black dashed line).

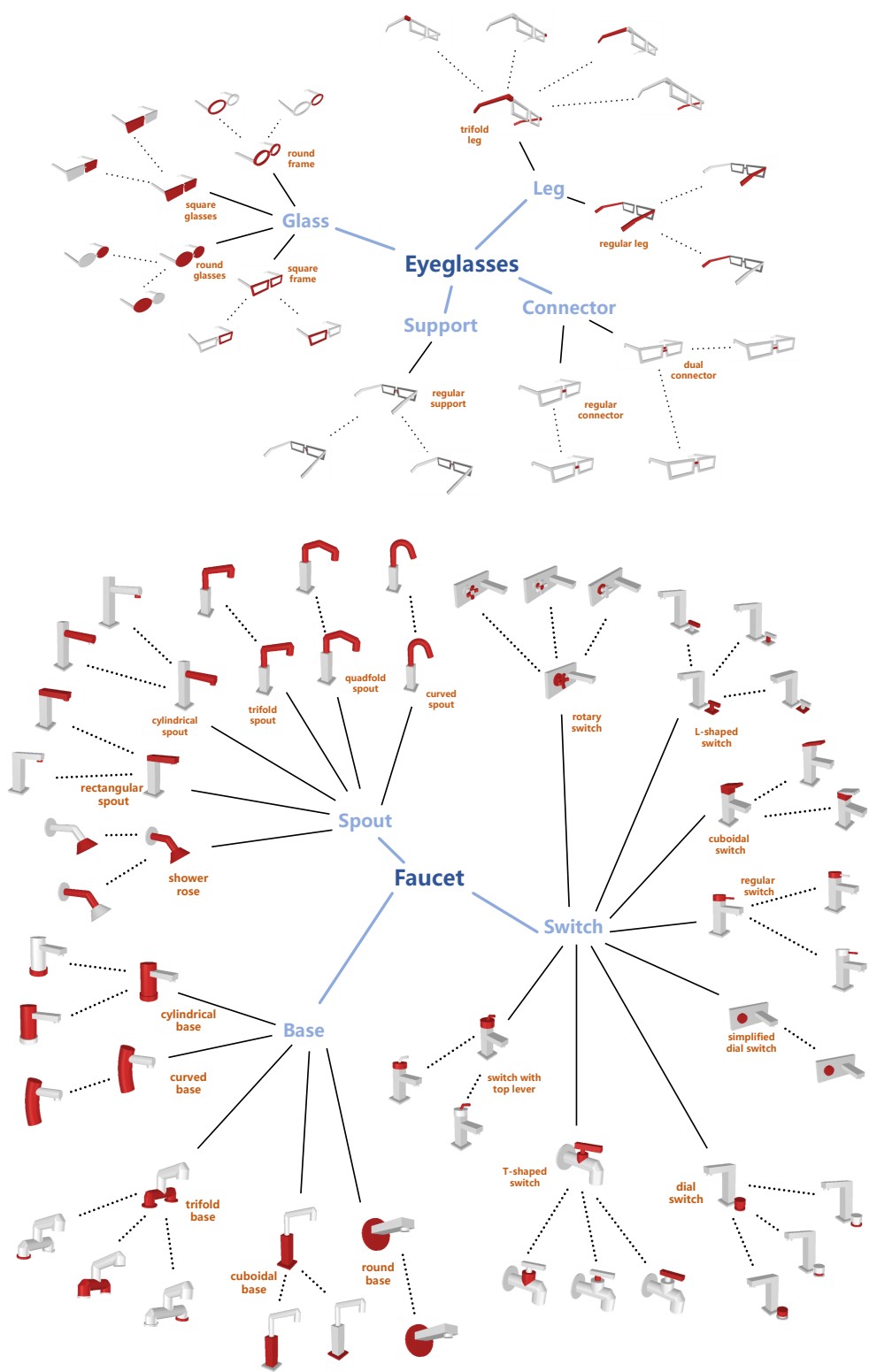

Figure 13: Visualizations for **Eyeglasses and Faucet** about i) how they are divided into a group of parts according to their structural for better guidance in the conceptualization process (blue solid line), ii) how they can be described and covered by a series of concept templates (black solid line), and iii) how concept templates are organized by geometry components (black dashed line).

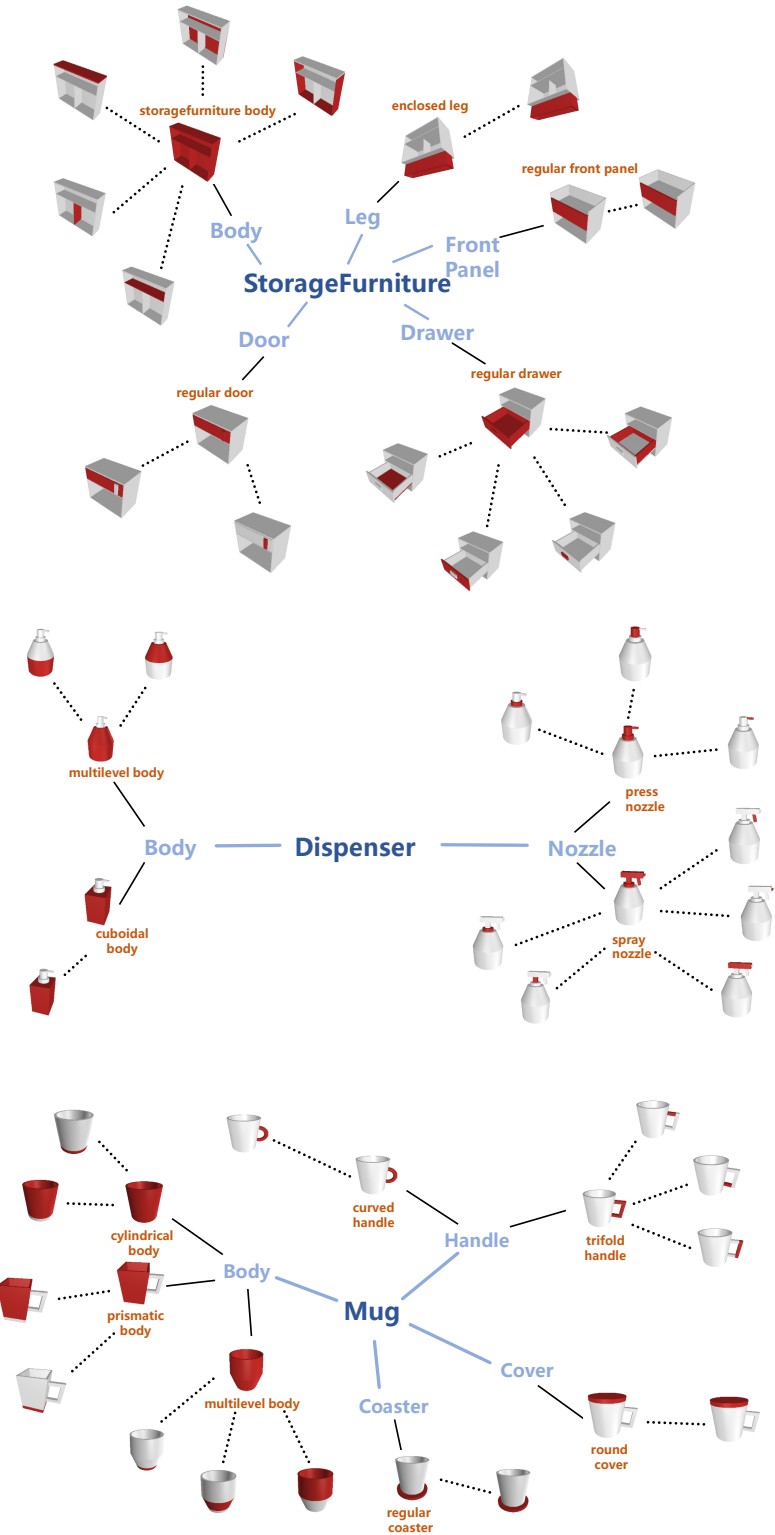

Figure 14: Visualizations for **StorageFurniture, Dispenser and Mug** about i) how they are divided into a group of parts according to their structural for better guidance in the conceptualization process (blue solid line), ii) how they can be described and covered by a series of concept templates (black solid line), and iii) how concept templates are organized by geometry components (black dashed line).

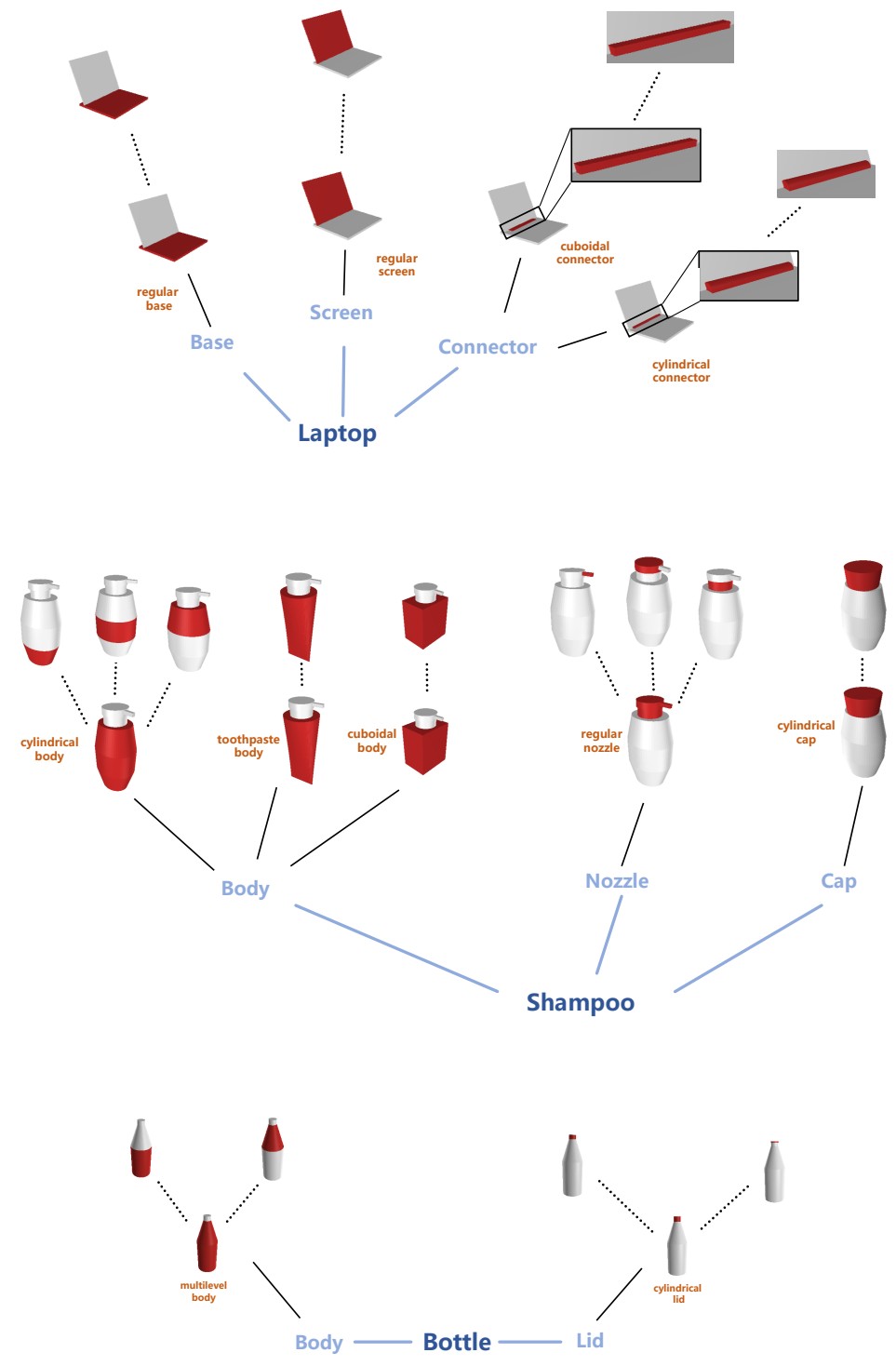

Figure 15: Visualizations for **Laptop, Shampoo and Bottle** about i) how they are divided into a group of parts according to their structural for better guidance in the conceptualization process (blue solid line), ii) how they can be described and covered by a series of concept templates (black solid line), and iii) how concept templates are organized by geometry components (black dashed line).

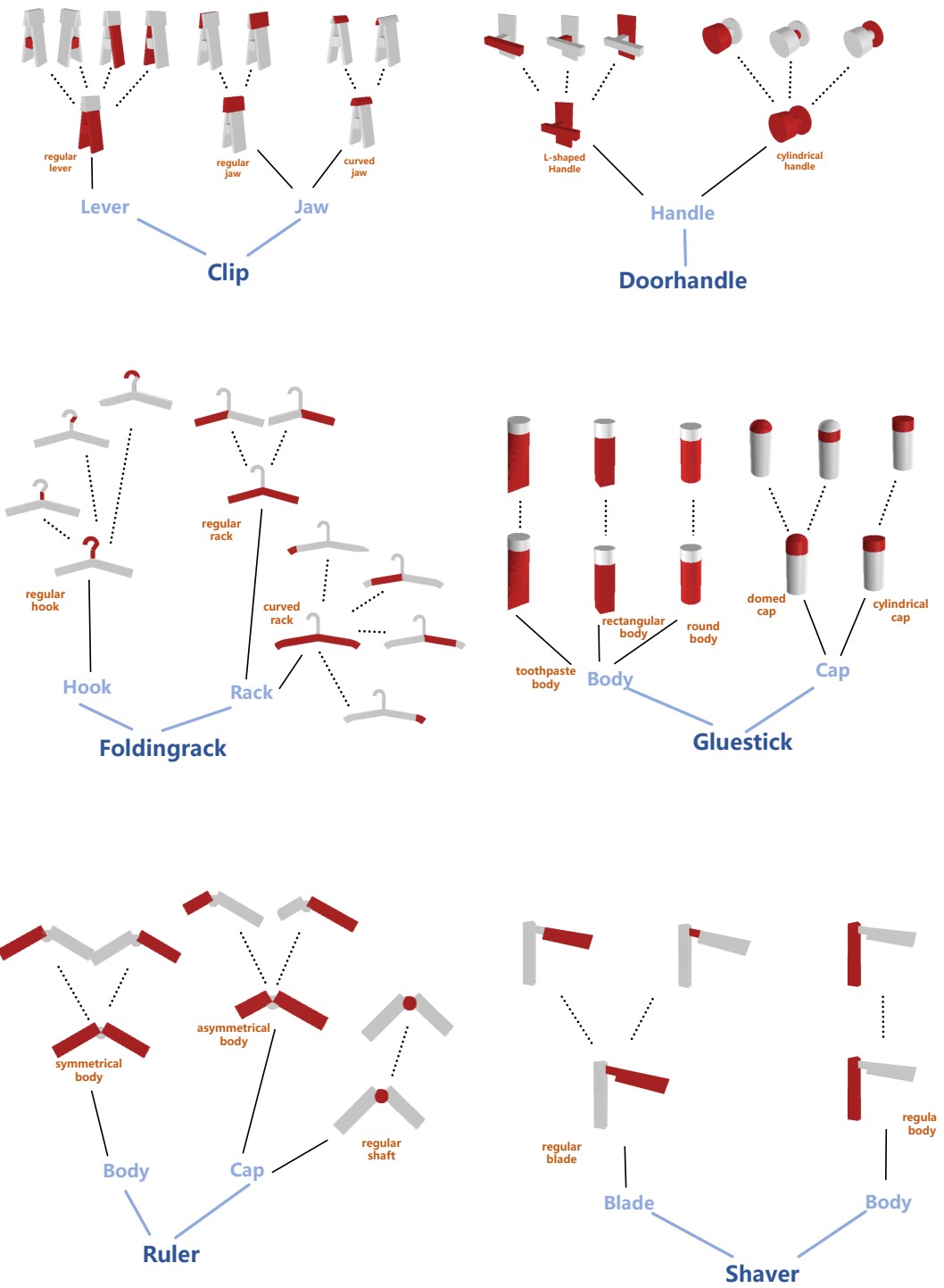

Figure 16: Visualizations for **Clip, Doorhandle, Foldingrack, Gluestick, Ruler and Shaver** about i) how they are divided into a group of parts according to their structural for better guidance in the conceptualization process (blue solid line), ii) how they can be described and covered by a series of concept templates (black solid line), and iii) how concept templates are organized by geometry components (black dashed line).