# OpenReview forum: "ConceptFactory: Facilitate 3D Object Knowledge Annotation with Object Conceptualization"
_NeurIPS.cc/2024/Datasets_and_Benchmarks_Track — NeurIPS 2024 Track Datasets and Benchmarks Poster_

### Official Review · Reviewer_Eyxu · 2024-07-25
**Toolbox and dataset for 3D object conceptualization**

**Rating:** 7
**Confidence:** 4
**Correctness:** Yes, the paper is correct.
**Clarity:** Yes, the paper is well written.

**Review:**

Strengths:
1. ConceptFactory may reduce the time and effort required to annotate complex 3D objects.
2. The method is rooted in human cognition theories about how humans perceive and conceptualize objects.
3. ConceptFactory has been successfully applied to different object categories, which is promising.
4. The paper introduces a large data set, which will be useful for other researchers.

Weaknesses:
1. ConceptFactory's performance could have been evaluated on more complex, novel or unusual objects.
2. Predefined templates may restrict the method's ability to generalize.
3. More user studies may be needed to understand how users interact with the system.

**Strengths:**

See above.

**Additional Feedback:**

None

**Documentation:**

Yes, sufficient detail on data collection, organization, and ethical use has been incorporated.

**Ethics:**

No ethical concerns.

**Limitations:**

Yes, the paper addresses limitations about the need for human annotation.

**Opportunities For Improvement:**

1. User studies that capture end user experience with the tool may help create an improved system.
2. The use of templates, while simplifying, may not generalize well.

**Relation To Prior Work:**

The prior work has been adequately discussed.

**Summary And Contributions:**

The paper introduces a method for annotating 3D objects by applying cognition theories to object conceptualization. The method, called ConceptFactory, includes a toolbox for annotation and a comprehensive dataset of conceptualized objects. The paper shows the effectiveness of this approach in different vision and robotics tasks.

---

> ### Author Rebuttal · Authors · 2024-08-17
>
> Thank you for the useful suggestions to improve our work! Our responses to the comments are detailed below.
>
> > **Q1: Evaluation on more complex, novel or unusual objects.**
>
> Thank you for the suggestion and we will search for more complex, novel or unusual objects to evaluate on to make our idea stronger. We also suggest that we have conducted evaluations on objects with real-world complexities. These objects are either real-world ones [41] or CAD models of real-world brands promoting products designed by experts [8].
>
> > **Q2: Predefined templates may restrict the method's ability to generalize.**
>
> Thank you for raising this concern and we respond with the following points. First, we find that our current template library can generalize well on objects with real-world level complexity [8,41] as suggested in Q1. Second, we suggest that since it is easy to customize additional templates to cover novel shapes in specific cases (L159-162), the generalization capability of our approach is extendable. We will also continue to design and release new templates to the community, which will further enhance our work's generalization capability.
>
>
> > **Q3: More user studies.**
>
> Thank you for the suggestion! We will conduct and add discussions on user studies in our revised version, and further use them to improve our conceptualization system.

---

### Official Review · Reviewer_mVG2 · 2024-07-26
**A toolbox for efficient annotation of 3D objects and a dataset of annotated objects for vision and robotic tasks.**

**Rating:** 6
**Confidence:** 4
**Correctness:** Yes, the claims are correct.
**Clarity:** The paper is well written but needs m…

**Review:**

The introduced tool is useful to annotate and understand objects and the dataset of objects is also validated by several vision and robotic tasks. However, the advantage of this annotation approach is not clear. Using this proposed complex annotation setting, the performance in the experiments is very similar to those using other annotation methods. Is the efficient annotation the only advantage introduced by this tool?

**Strengths:**

This work presents a useful tool as well as a large dataset for object understanding.

**Additional Feedback:**

No

**Documentation:**

Yes

**Ethics:**

I don't see major ethical concerns.

**Limitations:**

I didn't see potential negative societal impact.

**Opportunities For Improvement:**

The advantage of this annotation approach is not clear. It is better if more detailed comparison with other annotation methods could be provided. I also recommend showing more details about how to select categories and objects when building the dataset and the limitations of the current templates and knowledge settings.

**Relation To Prior Work:**

More comparison with previous annotation methods should be included.

**Summary And Contributions:**

This work presents a toolbox for efficiently annotating 3D objects based on a well-designed knowledge annotation paradigm. Templates are designed to describe geometry patterns and both region-based information and pose-based information can be annotated for object understanding. Using this tool, a dataset containing 5012 objects is generated and this dataset is used for vision tasks and manipulation tasks to show the effectiveness of the annotation approach.

---

> ### Author Rebuttal · Authors · 2024-08-17
>
> Thank you for the constructive feedback! Our responses to the comments are detailed below.
>
> > **Q1: The advantage of our annotation approach and comparison with previous annotation methods.**
>
> Thanks for this question! As suggested in L24-30 and L41-47, apart from being efficient, our approach can also provide **high-quality annotations for types that are extremely complicated to manually annotate previously**, which is another significant advantage of our approach.
>
> For the aspect of efficiency, L235-243 provides a discussion and comparison with previous annotation methods. Such efficiency can become more significant as more types of knowledge are annotated, and we consider the advantage on efficiency to be of great value.
>
> In addition, L289-300 demonstrates that our approach allows for human assignment of high-quality affordance annotations, which are acquired by interacting with objects in simulation in the previous annotation method [5] (L111). We conduct experiments on baseline methods Where2Act / Where2Explore trained with the affordance annotations acquired by [5,6] and our approach, and the comparison results in Tab.3 (particularly up to 26.6% improvement on push-Where2Act) clearly show the superiority of our affordance annotations in terms of both versatility and quality.
>
> We also provide a summary of our work's advantages in our general response, which are agreed upon by the reviewers. We will make our advantages and the comparisons clearer in our revision.
>
> > **Q2: Details on data selection and limitations.**
>
> Thanks for the suggestion! We have provided how we select objects in ConceptFactory asset in L248-250, and discussed about our limitations in Sec.A.2, Sec.A.3 of the supplementary material.
>
> For data selection, there are three major points. i) Widely used in vision and manipulation tasks. We refer to previous studies, *e.g.* [4,5,6] involved in our experiments, to select objects and categories to make sure that they can be well used in vision and manipulation tasks. ii) Including both CAD and scanned models. We mainly refer to popular data repositories [7,8,41] for both CAD and scanned models. iii) Yielding rich and diverse conceptualization results. We select 39 object categories including both simple ones like *Box* and complex ones like *Chair*, and finally achieve an average of 8.7 geometry instances and 29.0 parameters in each conceptualization result. For limitations, we will conduct user studies to find potential ones and further use them to improve our approach. We will add these additional details in our revision.

---

### Official Review · Reviewer_S87Q · 2024-07-28
**Review for the paper**

**Rating:** 6
**Confidence:** 4
**Correctness:** Yes
**Clarity:** Yes

**Review:**

The part about the paper that I find the most interesting is that objects can be decomposed as geometric concept templates and that properties about these templates that the object is composed off of can be propagated to the object. However, I have several questions -
1. The authors mention that objects of each category are divided into a group of parts according to their structure hierarchy for better guidance of the conceptualization process, but I fail to understand how would such a hierarchy or part-segregation/composition be available for novel objects in the first place for which this work of conceptualization would be most relevant?
2. As the concept templates are arbitrary, they don't necessarily represent meaningful object parts, which would be useful in tasks such as part-segmentation or pose estimation. How do the authors ensure that their predefined template concepts actually correspond to meaningful object parts?
3. This type of decomposition is not novel. Such decomposition is already followed during URDF creation where complex geometries are composed of more fundamental and simpler geometries.
4. I would have liked to understand the process of assigning templates closest to the object part's geometric structure as such a correspondence not only involves matching but also solving for pose and scale.

**Strengths:**

The paper is well-written and most parts can be well-understood. The major strength of this paper is in the proposal that complex geoemtries/objects can be defined as a composition of parts (predefined concept templates) for which several properties are already defined and can be propagated to the object. This allows for human effort to be involved only during the conceptualization process, and that properties related to several 3D tasks such as pose estimation, part-segmentation etc. can be annotated together.

**Additional Feedback:**

Written above.

Update: After reviewing the author's response, I have decided to raise my rating to 6 (marginally above acceptance threshold).

**Documentation:**

Yes

**Limitations:**

1. Would the 3D properties defined on template concepts translate to meaningful properties of actual parts. For instance, would the segmentation of a concept template correlate to segmenting a meaningful part of the object.
2. The formulation about object part decomposition and conceptualization is not entirely novel. Existing 3D definition files such as URDF already define the 3D structure of complex geometries as a collection of simpler/fundamental geometries (that includes parameterizing their scale, pose, etc. and by extension segmentation).
3. How expensive would the approach be for conceptualizing and annotating objects that don't have predefined structural hierarchy? (object's parts)

**Opportunities For Improvement:**

I see that the conceptualization and annotations were performed from relatively well-known 3D scans/objects. I would have liked to see such conceptualization being performed for some real scans/objects.

**Relation To Prior Work:**

Yes

**Summary And Contributions:**

The work proposes a method/system for efficiently annotating a 3D object's knowledge by recognizing that object through generalized concepts. The paper proposes Conceptfactory Suite - which is a web based interface that enables object conceptualization and Conceptfactory Asset - which is a large collection of conceptualized objects acquired through the Conceptfactory Suite. The authors propose that efficient annotation of 3d object knowledge is useful for various 3d tasks such as object/part segmentation, grasping, object/part based pose estimation etc. The main advantage of this approach is that knowledge on the generalized concept (concept template) can be propagated to the object and that human effort is required only once during the conceptualization phase.

---

> ### Author Rebuttal · Authors · 2024-08-17
>
> Thank you for the valuable comments! Since we find some of the questions are related to the conceptualization process, we suggest that the video demo provided in our supplementary material **"*conceptualization_platform/platform_demo*" may be very helpful for better understanding of this point**. We provide our responses for each question below and look forward to further discussions if there are still any unclear points.
>
> > **Q1: Obtaining hierarchies or part-segregation/composition for novel objects.**
>
> To avoid miscommunication, we first respectfully provide our understanding of the question that the reviewer thinks there is **a process** in our workflow to *divide an object's whole mesh into multiple part meshes according to a kind of part annotation (hierarchy or part-segregation/composition)*. Under this assumption, we respond to the question as follows.
>
> We would like to clarify that **the above process does not exist in our workflow**. The **structural hierarchy** in our paper is *not a kind of part annotation to segregate the part meshes from an object mesh*, but **a tree diagram for each object *category* as shown in Supp-Fig.5-17** (solid line, refer to the caption for more details). Therefore, referring to the whole paragraph in L169-185 and our demonstration video "conceptualization_platform/platform_demo", what we actually do is that **we separate the conceptualization workflow of an object into several steps guided by the structural hierarchy tree diagrams for different categories, and in each step a concept template for a specific part semantic in the tree diagram can be selected and parameterized by a conceptualization annotator**. The conceptualization workflow does not rely on any part annotations or involve any operations to divide the object mesh.
>
> The video demo provided in our supplementary material "**conceptualization_platform/platform_demo**" clearly shows this point.
>
> Since a structural hierarchy tree diagram is for **a specific object category**, it is available for novel objects in this category. And it is also straightforward to define new structural hierarchy tree diagrams for novel categories. We thank the reviewer for bringing up this point that could potentially cause confusion. We will carefully rephrase the paragraph (L169-185) to make this point clear and avoid any confusion in our final version.
>
> > **Q2: Correspondence between concept templates and meaningful object parts.**
>
> We thank the reviewer for raising this question. When defining concept templates, we ensure that each concept template and its geometry components correspond to meaningful **part semantics**. Brief illustrations of this point are in Supp-Fig.5-17 (black line). During the conceptualization process, the conceptualization annotator is asked to select concept templates and parameterize them to effectively approximate the geometric structure of each specific meaningful part of the object. By completing this task, the correspondence between a concept template instance and a meaningful object part can be established. We will make this point clearer in the revised version.
>
> > **Q3: Novelty.**
>
> Thanks! We respectfully clarify that our true novelty lies in __developing a concept-driven annotation method (*i.e.* the ConceptFactory annotation system) to acquire different types of 3D object annotations (*e.g.* part annotations, affordance annotations, *etc.*) for the first time__, rather than proposing a description format similar to URDF.
>
> This essence can be revealed by a remarkable property of our approach that our concept templates enable mathematical definition of various types of object knowledge and automatic knowledge annotation propagation to an object once the object is conceptualized (L215-219). This greatly distinguishes our work (an annotation method) from the specific description format URDF, in which the links are typically represented by meshes and is unable to be used for flexible and efficient knowledge annotation.
>
> We will make this point clearer in our revision, and more of the novelties and contributions of our work, which are agreed upon by other reviewers, can be referenced in our general response.
>
> > **Q4: Process of assigning templates closest to the object part's geometric structure.**
>
> During conceptualization, the conceptualization annotator first chooses a template that best matches the part's geometric structure (L177-178), and then the chosen template is parameterized for effective approximation (L179-181). The parameterization process also involves finding the right pose parameters for the template instance, whereas the scale is implicitly determined by some template parameters (*e.g.* spheres with diameters of different values have different scales). We will make this point clearer in the revised version.
>
> > **Q5: Conceptualization performed for real scans/objects.**
>
> Thank you for the suggestion. Many objects we have conceptualized are either real-world ones [41] or CAD models of real-world brands promoting products designed by experts [8]. We will also follow the suggestion to collect our own real scans/objects and test our approach on them to make our idea stronger.
>
> > **Q6: Cost for conceptualizing and annotating objects that don't have predefined structural hierarchy.**
>
> Thanks for the question. According to our clarification in Q1, our structural hierarchy tree diagram is for a category of objects. Therefore, we respectfully suggest that this question may be caused by a misconception.

---

> > ### Comment · Reviewer_S87Q · 2024-08-18
> > **Response to Rebuttal**
> >
> > I thank the authors for their response to my review - all my questions have been addresed.
> >
> > I have looked at the video now, and my questions regarding the object conceptualization workflow have been clarified. Given this, I would like to increase my score to 6 (Marginally above acceptance threshold).

---

> > > ### Author Response · Authors · 2024-08-19
> > >
> > > Thank you very much for your reply! We are delighted to find our response helpful and sincerely express our gratitude for increasing the score to 6 (Marginally above acceptance threshold). This is of great help and significant meaning to us!
> > >
> > > However, we have noticed that the score still shows as 5 on OpenReview. Therefore, we would like to politely inquire if the modification may not be successfully submitted?
> > >
> > > We are looking forward to this update!

---

### Official Review · Reviewer_r1o8 · 2024-08-07
**Solid work on 3D object knowledge annotation**

**Rating:** 6
**Confidence:** 4
**Correctness:** Yes.
**Clarity:** Yes.

**Review:**

This work presents a high-quality and original approach to 3D object knowledge annotation, leveraging human cognitive theories to improve efficiency and versatility. The clarity of the paper is commendable, effectively explaining the innovative ConceptFactory framework and its components. This work has great potential to transform both vision and robotics tasks by significantly reducing the manual effort required for detailed object annotations.

**Strengths:**

- The paper introduces a novel approach to 3D object knowledge annotation that addresses the limitations of traditional methods, such as the labor-intensive and time-consuming nature of manual annotations.
- The ConceptFactory framework significantly reduces the time and human effort required for annotating 3D objects by enabling automatic propagation of various types of knowledge once an object is conceptualized.
- The ConceptFactory Suite includes a Standard Concept Template Library and a web-based platform, providing a unified and user-friendly toolbox for object conceptualization and annotation.
- The ConceptFactory Asset offers a large collection of pre-conceptualized objects, facilitating research and reducing the need for extensive manual annotation efforts.
- The approach is inspired by the 'Recognition-by-Components' theory in human cognition, which adds a theoretical foundation and aligns the annotation process with human perceptual mechanisms.

**Additional Feedback:**

It would be helpful to share an online demo and a small sample of your dataset.

**Documentation:**

I didn't find any code, although the videos in the supplementary materials look promising.

**Ethics:**

No other ethical concerns.

**Limitations:**

Yes, the limitations are discussed in the supplementary materials.

**Opportunities For Improvement:**

- The approach may struggle with highly complex and irregular objects that do not fit neatly into predefined geometric concepts.
- It seems that customizing and extending concept templates for novel shapes might require some effort.
- The framework might still face challenges with highly intricate and subjective annotations requiring fine-grained semantic understanding.

**Relation To Prior Work:**

Yes.

**Summary And Contributions:**

This paper introduces ConceptFactory, a framework designed to improve the efficiency of annotating 3D object knowledge by recognizing 3D objects through generalized concepts. It consists of two main components: ConceptFactory Suite and ConceptFactory Asset. The system facilitates efficient acquisition and customization of extensive object knowledge, validated through various benchmark tasks.

# Contributions
- ConceptFactory introduces a new paradigm for 3D object knowledge annotation, addressing the complexities and labor-intensive nature of traditional methods.
- ConceptFactory Suite: This suite includes a comprehensive toolbox and a web-based platform that uses a Standard Concept Template Library to describe objects through generalized geometric concepts.
- ConceptFactory Asset: A large dataset of conceptualized objects, offering a valuable resource for researchers to study different object understanding tasks without the need for extensive manual annotations.
- The effectiveness of ConceptFactory is demonstrated on a range of vision and robotics tasks, showcasing the high quality and versatility of the annotations.

---

> ### Author Rebuttal · Authors · 2024-08-17
>
> Thank you for the detailed and insightful feedback! Our responses to the comments are listed below.
>
> > **Q1: The approach may struggle with highly complex and irregular objects that do not fit neatly into predefined geometric concepts.**
>
> We thank the reviewer for the insightful viewpoint that objects that are more complex and irregular tend to be harder to fit into the predefined templates, and we respond from two perspectives.
>
> First, we find that the representational power of our proposed concept template library has been tested through our conceptualizations on objects with real-world level complexity. In fact, many objects we have conceptualized are from either scanning real-world objects (objects from AKB-48) or [3D Warehouse](https://3dwarehouse.sketchup.com/) (objects from PartNet-Mobility), which is a 3D model library containing CAD models of real-world brands promoting products designed by experts. Some illustrations are provided in the rebuttal PDF.
>
> Second, we would like to suggest that it is easy to design and customize additional templates to better adopt to potentially more complex objects, rather than being confined to "predefined" ones (L159-162).
> We will also continue to design and release new templates to further enhance our work's capability in covering more complex objects.
>
>
> > **Q2: It seems that customizing and extending concept templates for novel shapes might require some effort.**
>
> Thanks for this question! We first highlight that thanks to the inheritable nature of a template representation, users can easily customize new templates using existing ones to cover novel shapes in specific applications (L159-162). We also suggest that considering the significant advantage of ConceptFactory over conventional annotation scheme in terms of annotation efficiency (L235-243), the effort required for such customization is preferable. Further, as we have promised in the previous question, we will continue to design and release new templates to further extend ConceptFactory to more novel shapes and categories for the community, which can reduce the users' effort in this aspect.
>
>
> > **Q3: The framework might still face challenges with highly intricate and subjective annotations requiring fine-grained semantic understanding.**
>
> Thanks! We provide examples in the rebuttal PDF to show the capability of our framework to give fine-grained knowledge annotations and will add them into our revised version.
>
>
> > **Q4: Online demo and sample dataset.**
>
> Thank you for the suggestion! A link to the Google Drive containing a sample dataset is in L393 of the supplementary PDF. We will develop an online demo and share our codes with the community.

---

### Author Rebuttal · Authors · 2024-08-17

We sincerely thank the reviewers for their time devoted to reviewing our submission. Among the reviews, we are very delighted to see the reviewers' recognition of our work, finding the following strengths:

- Our work introduces a well-designed new paradigm for 3D object knowledge annotation, including a unified and user-friendly toolbox for object conceptualization and annotation (Reviewer `r1o8`, `mVG2`).

- Our work is rooted in human cognition theories about how humans perceive and conceptualize objects, and aligns the annotation process with human perceptual mechanisms (Reviewer `r1o8`, `Eyxu`).

- Our work addresses the complexities and labor-intensive nature of traditional methods (Reviewer `r1o8`) and significantly reduces the time and human effort required for annotating 3D objects (Reviewer `r1o8`, `S87Q`, `Eyxu`).

- Our work offers a large collection of object conceptualizations (Reviewer `r1o8`, `mVG2`, `Eyxu`), which is a valuable resource for researchers to study different object understanding tasks without the need for extensive manual annotations, facilitating research and reducing the need for extensive manual annotation efforts (Reviewer `r1o8`, `Eyxu`).

- Our work has been successfully applied to different object categories (Reviewer `Eyxu`), and it is effective in a range of vision and robotics tasks, showcasing the high quality and versatility of the annotations (Reviewer `r1o8`).

We are also happy to know that the reviewers are satisfied with the correctness and clarity of our paper, and will respond to each reviewer in their respective rebuttal sections.

---

### Decision · Program_Chairs · 2024-09-26

**Decision:**

Accept (Poster)

**Comment:**

The paper proposes to model 3D objects using standard concept templates, where objects are represented by parts composed of parameterized  collection of geometric primitives.  To allow objects to be modeled using concept templates, a platform is introduced that 1) allows users to select different concept templates for different parts, 2) automatically adjust the template parameters so that the conceptualized object matches the original object geometry, 3) annotation of semantics and pose.   Using this process, a dataset of 5012 objects covering 39 categories is provided.  Experiments are conducted on various tasks (segmentation, pose estimation, manipuation).

Overall, all four reviewers were positive on this work.  The reviewers found the paper to be well-written, and the decomposition of objects into well-defined templates to be useful for object understanding.

The AC agrees the work as value and recommends acceptance.  The authors are encouraged to improve the paper based on reviewer feedback.  In particular, the authors should provide user study, discussion about the limitations of the approach, ability to handle objects with complex geometry, and clarify details based on reviewer comments.